**RESEARCH**

# Cell type determination for cardiac differentiation occurs soon after seeding of human-induced pluripotent stem cells

Connie L. Jiang[1], Yogesh Goyal[2,3,4], Naveen Jain[1], Qiaohong Wang[5,6], Rachel E. Truitt[5,7], Allison J. Coté[8], Benjamin Emert[9], Ian A. Mellis[9], Karun Kiani[1], Wenli Yang[5,7,10], Rajan Jain[5,6,7,10,11*] and Arjun Raj[4,11*]

*Correspondence:
jainr@pennmedicine.upenn.
edu; arjunraj@seas.upenn.
edu
[11] Department of Genetics,
Perelman School of Medicine,
University of Pennsylvania,
Philadelphia, PA, USA
Full list of author information
is available at the end of the
article

## Abstract

**Background:** Cardiac differentiation of human-induced pluripotent stem (hiPS) cells consistently produces a mixed population of cardiomyocytes and non-cardiac cell types, even when using well-characterized protocols. We sought to determine whether different cell types might result from intrinsic differences in hiPS cells prior to the onset of differentiation.

**Results:** By associating individual differentiated cells that share a common hiPS cell precursor, we tested whether expression variability is predetermined from the hiPS cell state. In a single experiment, cells that shared a progenitor were more transcriptionally similar to each other than to other cells in the differentiated population. However, when the same hiPS cells were differentiated in parallel, we did not observe high transcriptional similarity across differentiations. Additionally, we found that substantial cell death occurs during differentiation in a manner that suggested all cells were equally likely to survive or die, suggesting that there is no intrinsic selection bias for cells descended from particular hiPS cell progenitors. We thus wondered how cells grow spatially during differentiation, so we labeled cells by expression of marker genes and found that cells expressing the same marker tended to occur in patches. Our results suggest that cell type determination across multiple cell types, once initiated, is maintained in a cell-autonomous manner for multiple divisions.

**Conclusions:** Altogether, our results show that while substantial heterogeneity exists in the initial hiPS cell population, it is not responsible for the variability observed in differentiated outcomes; instead, factors specifying the various cell types likely act during a window that begins shortly after the seeding of hiPS cells for differentiation.

## Background

Differentiation of human-induced pluripotent stem (hiPS) cells is highly variable even when using standardized protocols and produces heterogeneous samples that are unfit for regenerative medicine or research models without further selection. Even

within clonal cell lines, differentiation outcomes can be highly variable, pointing to a role for non-genetic factors in cell type determination [1–5]. We call a cell "determined" if, barring any further environmental change, it will ultimately differentiate to a particular cell type. It remains unknown when in the differentiation process (during or even before) these non-genetic factors act to cause cells to adopt a particular cell type.

An important example is the differentiation of hiPS cells into cardiac tissue. Numerous groups have developed and optimized differentiation protocols, leading to vast improvements in yield and efficiency of cardiomyocyte generation. However, protocols still consistently produce a mixed population of cardiomyocytes and non-cardiomyocytes, leading many groups to attempt to isolate cardiomyocytes post-facto with metabolic selection or cell sorting [2, 6–10]. Additionally, substantial cell death concomitant with cell growth is routinely observed during differentiation, and it is not known whether certain cells are predetermined to survive [11]. In vivo evidence suggests that multipotent precursors that generate a variety of cardiac cells, including cardiomyocytes, vascular endothelial cells, and smooth muscle cells (but not cardiac fibroblasts), choose a fate at a point prior to gastrulation [12–16]. That timing of cell type determination is earlier than would have been expected from previous markers proposed based on in vitro differentiation of cultured hiPS cells; hence, the timing of determination to these different cell types during directed differentiation remains unknown and may be earlier than has been previously suggested [4, 17–22].

It has been suggested that the ultimate heterogeneity in cell types may result from intrinsic differences in pluripotent stem cells even before differentiation begins [23–25]. Along these lines, recent results challenge the traditional notion that all pluripotent cells have the same developmental potential. Even among human embryonic stem cells that possess equivalent levels of pluripotency markers like Oct4, interconvertible subpopulations identified by differential expression of other marker genes predicted biased differentiation toward distinct cell types [23, 26–29]. At the same time, the proportion of differentiated cell types resulting from stem cells initially found in these different expression states (defined as the levels of expression of all genes in a cell) was shown to be strongly modulated by extrinsic cues such as stem cell culture media and cell-matrix adhesion substrate [29–32]. Hence, it remains unresolved whether certain hiPS cells are intrinsically primed to become specific cell types following cardiac differentiation, or if cell type determination occurs later along the path of differentiation following extrinsic cues.

One way to answer this question is to longitudinally track cells and their fates through the differentiation process. Single-cell transcriptome profiling has revealed considerable variability in the expression states of both human pluripotent stem cells and the cells that result from their directed differentiation into cardiomyocytes [33–36]. However, it is difficult to know at what point a newly apparent transcriptomic signature signifies a determination event. The use of DNA barcodes combined with single-cell RNA sequencing to track the clonal history and expression state of a cell simultaneously [37, 38] enables one to map heterogeneity over time to specific clonal populations. Such techniques can reveal when putative cell type determination events occur by combining barcoding with experimental designs that ask

whether "identical twin" cells adopt the same cellular states under various conditions. However, such experiments have not been performed in the context of cardiac differentiation.

Here, by connecting individual cardiac differentiated cells that share a common hiPS cell progenitor, we show that, although within a single differentiation experiment cells that share an hiPS cell progenitor tend to adopt more similar expression states than expected by chance, cells sharing an hiPS cell progenitor that are seeded across parallel differentiations do not adopt similar cell types. Moreover, cells all have an equal chance to survive cardiomyocyte differentiation regardless of whether or not they share an hiPS cell progenitor, suggesting no intrinsic bias for survival versus death. Taken together, these results suggest that final cell type determination has not yet occurred in the hiPS cell population but instead occurs once the differentiation process is underway.

## Results

### Single-cell RNA sequencing following cardiomyocyte directed differentiation of hiPS cells reveals an increase in the heterogeneity of cellular expression states

We wanted to measure the extent to which different expression states arise after differentiating genetically identical hiPS cells. We used a well-established cardiac differentiation protocol [39–44] that uses small molecules and growth factors to efficiently push cells through stage-specific transitions as they become cardiomyocytes (Fig. 1A). We used PENN123i-SV20 hiPS cells as the parental cell line for this study [45] and chose to transcriptionally profile single cells on day 14 or 15 of differentiation, shortly after we began to observe contractile activity and cardiomyocyte-specific gene expression. While many groups metabolically select for cardiomyocytes at this stage with glucose-depleted and/or lactate-enriched media [2, 6–10], we chose to forgo this step as we wanted to preserve the non-cardiomyocyte cell types that may otherwise be lost. Following quality control analysis, we captured a total of 17,599 differentiated cells with droplet-based single-cell RNA sequencing (10X Chromium platform v3), from which we captured the expression of 24,728 genes in 17,599 cells. We used Seurat v4 [46] to cluster cells by expression similarity into 15 clusters and used Uniform Manifold Approximation and Projection (UMAP) to visualize the transcriptional differences between these cells (Fig. 1B). Using canonical marker genes, we were able to attribute putative cell types to several of these clusters, including fibroblasts, cardiomyocytes, cardiac progenitors, epicardium, or epithelium (Fig. 1C, Additional file 1: Fig. S1A).

Seeing the heterogeneity in gene expression across individual cells following differentiation, we wondered how this level of variability compared with variability in the initial hiPS cell population. After appropriate normalization and variance stabilization (see "Methods"), we performed principal component analysis on both our differentiated cell dataset as well as a separate dataset consisting of 1198 undifferentiated hiPS cells. In a differentiated population consisting only of one final cell type and its intermediates, we would expect a single primary axis of variation associated with cell type maturity. In this hypothetical case, there may only be one "significant" principal component explaining the majority of the variance in that system. In contrast, a highly variable population consisting of many cell types would be likely to have multiple axes of variation, each explaining some non-negligible proportion of the total variance. We found that the dropoff in

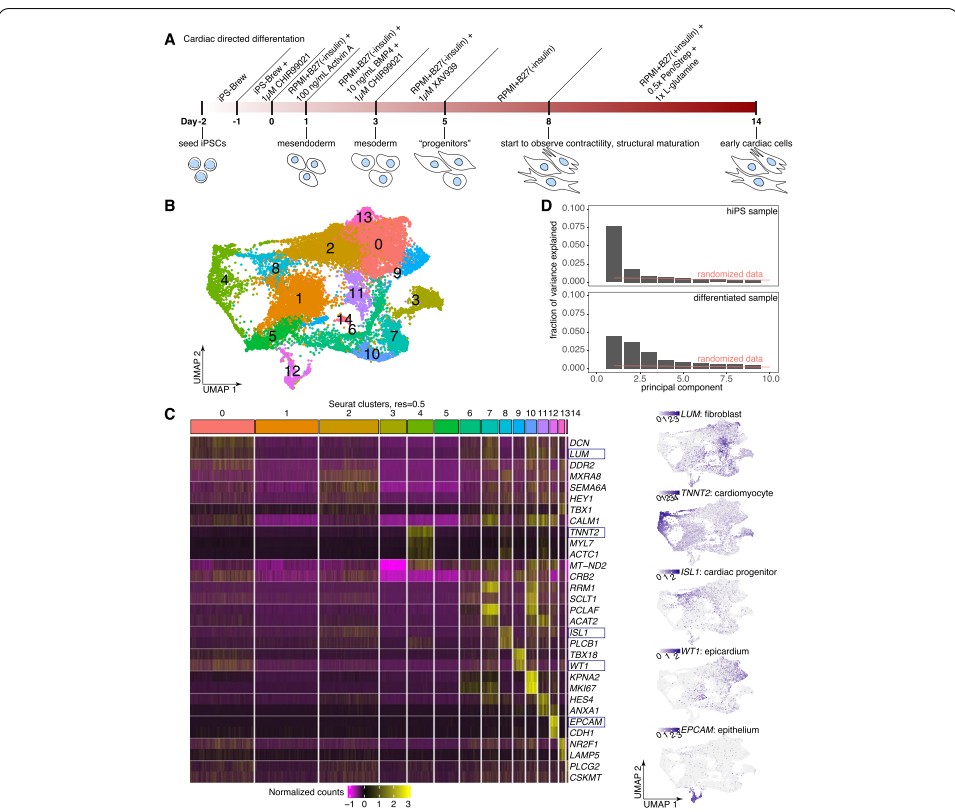

**Fig. 1** Cardiac directed differentiation of hiPS cells results in extensive heterogeneity of cell expression states. **a** We directed hiPS cells to differentiate toward cardiac cell types using a well-established monolayer small molecule protocol. **b** We performed single-cell RNA sequencing on day 14 differentiated cells (*n* = 17,599 in total after filtering). We applied the UMAP algorithm to the first 50 principal components to visualize differences in gene expression. Cells are colored by clusters determined using Seurat's *FindClusters* command at a resolution of 0.5 (i.e., "Seurat clusters, resolution = 0.5"). **c** Heatmap showing normalized gene expression for 2–3 selected markers per Seurat cluster across all 17,599 cells. Maintaining the organization provided by UMAP, we recolored each cell by its expression of *LUM*, a fibroblast marker found largely in Seurat clusters 0–2, *TNNT2*, which marks the putative cardiomyocytes in Seurat cluster 4, *ISL1*, which marks cardiac progenitors in Seurat cluster 8, *WT1*, which marks epicardial cells in Seurat cluster 9, and *EPCAM*, which marks epithelial cells in Seurat cluster 12. **d** Fraction of variance explained by each of the top 10 principal components for an hiPS cell single-cell RNA sequencing dataset (*n* = 1,198, top) and for the day 14 differentiated cell dataset. The red line indicates how much variance is explained when the data is randomized prior to PCA (i.e., noise, see "Methods")

fraction of variance explained per principal component was steeper for the hiPS cell sample as compared to the differentiated sample (i.e., fewer principal components that were distinguishable from randomized data), suggesting that through directed differentiation, the differentiated population became more variable than their initial states (Fig. 1D).

## Related hiPS cells sometimes show similar expression states upon cardiac directed differentiation

The fact that we observed an increase in the variability of gene expression states as hiPS cells differentiate toward cardiac cell types led us to wonder, how are the progeny of individual hiPS cells distributed across differentiated cell expression states? One

possibility is that cell type determination occurs early in or even before differentiation and is then maintained. In that case, differentiated clones (i.e., sets of cells that share an hiPS cell precursor) would tend to show similar expression states following cardiac directed differentiation. At the other extreme, cell type determination may occur so late in differentiation, or fail to be maintained through cell divisions, such that related cells belonging to a single clone would be found randomly distributed across all possible differentiated expression states. (Such random, late determination could be due either to variable intrinsic factors that lead to differences between related cells in determination or to microenvironmental differences that instruct related cells to adopt different expression states.) While single-cell RNA sequencing alone is agnostic to relationships between cells, combining it with lentiviral barcoding of individual hiPS cells and their progeny enabled us to track the differentiated expression states associated with particular hiPS cells. We thus were able to ask: are differentiated cells derived from the same initial barcoded hiPS cell more similar to each other in expression state than to the differentiated population as a whole?

We followed the example of previous methods that use lentivirally integrated DNA barcodes to mark cells that are descended from a common progenitor [37, 38]. These barcodes lie in the 3′ untranslated region of a reporter transgene such that they are transcribed by the target cell and all its progeny and can be captured by single-cell RNA sequencing. We previously generated such a barcode library for use with Rewind, a method that allows for the direct profiling of rare cells of interest through the combination of genetic barcoding with RNA fluorescence in situ hybridization [47]. Using this barcode library, which consists of random 100-mers in the 3′ untranslated region of a green fluorescent protein (GFP) transgene, we transduced hiPS cells prior to seeding for cardiac differentiation. We harvested the differentiated cells 16 days later, sorted for GFP-positive cells to enrich specifically for those that expressed a lentiviral barcode, and sequenced the GFP-positive cells (Fig. 2A). As transcripts of the GFP transgene are captured by single-cell RNA sequencing, we were able to recover and connect Rewind barcodes to individual cells through analysis of GFP transcript 10× sequencing reads (Additional file 1: Fig. S2-3). We also profiled some of our GFP-negative cell population and found the populations to be very similar to the GFP-positive populations (Additional file 1: Fig. S2), suggesting that the introduction of the barcode itself did not introduce major bias (there was some bias in clusters 1 and 3, but those clusters were difficult to interpret biologically due to a lack of good markers and hence were not subjected to further analysis (Additional file 1: Fig. S3C)). Through this process, we were able to both profile the expression state of individual differentiated cells and determine which differentiated cells were descended from the same barcoded hiPS cell progenitor.

We wondered whether differentiated cells deriving from the same barcoded hiPS cell would be more transcriptionally similar to each other than to the differentiated population at large. To address this question, we analyzed the 49 barcode clones consisting of at least 20 cells following cardiac directed differentiations (consisting of two replicate differentiations, with 30 and 19 barcode clones respectively). We asked whether differentiated cells that shared a barcode were more clustered in gene expression space than a matched number of cells randomly sampled from the same pool of cells labeled by these 30 or 19 barcodes. As a proxy for gene expression space, we used the Seurat cluster

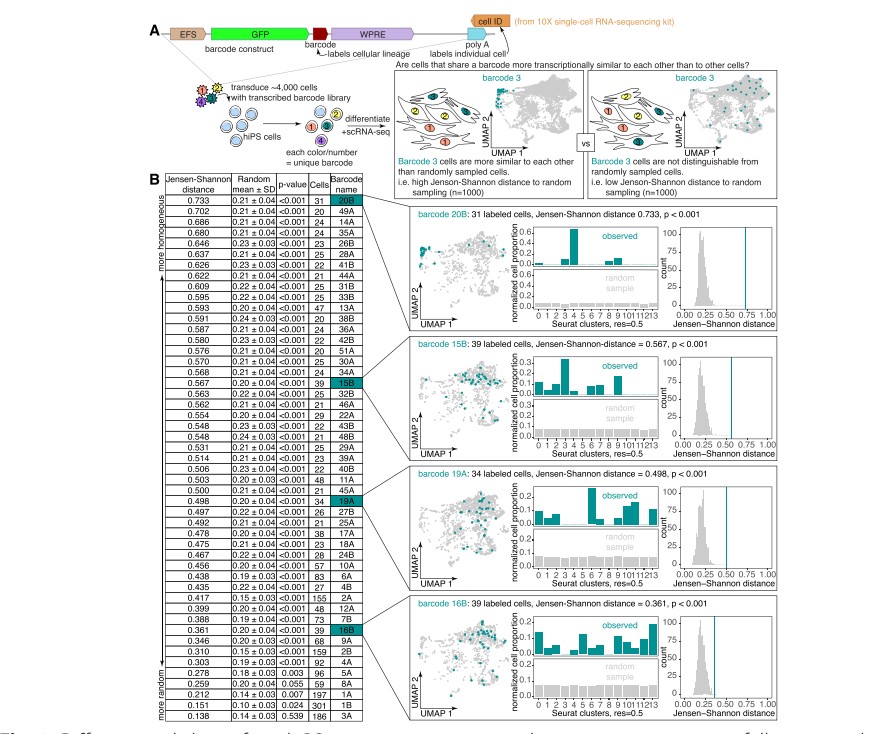

**Fig. 2** Differentiated clones from hiPS precursors sometimes cluster in expression state following cardiac differentiation. **a** Schematic of barcoding approach for labeling hiPS cells prior to cardiac differentiation. We transduced 4000–8000 hiPS cells at an MOI of ~0.03 with the Rewind [47] barcode library. After 3 days (3–4 population doublings), we seeded cells for cardiac differentiation. On day 14 of cardiac differentiation, we performed single-cell RNA sequencing. We asked whether cells sharing a barcode were more transcriptionally similar on day 14 than randomly sampled barcoded cells. **b** Table of all 49 barcode clones comprising 20 or more cells within a differentiation, ranked by homogeneity of expression states of constituent cells, as measured using the "Jensen-Shannon distance", i.e., the observed Jensen-Shannon distance between the cluster probability distribution associated with each barcode and a random cluster probability distribution generated from 1000 samples of a matched number of cells. "Random mean +/− SD" refers to the mean and standard deviation of randomized Jensen-Shannon distances (see "Methods"), which were used to calculate the "*p*-value" column. "Cells" refers to the number of constituent cells per barcode. Pullout plots for 4 of the profiled barcodes (teal background) labeling 30–40 constituent cells, exhibiting a range of Jensen-Shannon distances (suggesting Jensen-Shannon distance is not a product of cell number) are described as follows. Maintaining the organization provided by UMAP, we plotted all cells in the analysis (gray) and recolored cells corresponding to each featured barcode in teal. We also plotted bar graphs for observed cluster probability distribution (teal) and the average random cluster probability distribution (gray). Finally, we plotted histograms demonstrating the distribution of randomized Jensen-Shannon distances between random cluster probability distributions generated from 1000 additional random samples and the average random cluster probability distribution described above (gray) versus the observed Jensen-Shannon distance between the featured barcode cluster probability and the average random cluster probability distribution (teal vertical line)

assignments described in Fig. 1B. Since the clusters contain different numbers of cells, and different proportions of cells in each cluster are barcoded, we calculated the distribution of each barcode across clusters, accounting for differences in cluster size and barcode diversity (see "Methods") to generate "cluster probability distributions" for each barcoded clone (Additional file 1: Fig. S4A).

As a metric of similarity, we chose to use the Jensen-Shannon distance, an entropy-based measurement that has been used previously to calculate cell type or tissue-specificity scores [48, 49]. The Jensen-Shannon distance metric [50] quantifies the relative

distance between 2 probability distributions (not assuming any ordinality to the individual bins), where a distance of 0 denotes 2 identical distributions and a distance of 1 denotes maximally different distributions. For each of the 49 barcodes, we averaged 1000 random samples of a matched number of cells, generating a roughly uniform distribution of normalized proportions, and then queried the Jensen-Shannon distance between the observed barcode cluster probability distribution as compared to the average randomly sampled distribution (Fig. 2B). To determine the significance of these Jensen-Shannon distance values, we also calculated the Jensen-Shannon distance for 1000 additional random samples of a matched number of cells. We found that for all but 2 barcodes, the calculated Jensen-Shannon distance from the observed barcode distribution to the random distribution was significantly ($p < 0.05$) greater than what we might expect from random chance, suggesting that cells that share a barcode tend to cluster more than would be seen by random chance.

Even so, the degree to which final expression states were similar between cells within a clone varied from barcode to barcode. Sometimes cells belonging to a barcode clone predominantly localized to a single specific cluster (ex. barcode B20 in cluster 4), while other times, a barcoded clone had a broad distribution across many clusters, although its constituent cell expression states were still more similar than random (ex. barcode B1). We wondered if certain clusters had a greater propensity to encompass all the cells within a clone descended from a given barcoded hiPS cell than others. We found that the expression clusters with the strongest tendency to encompass all of a barcoded clone were clusters 4, 8, 3, and 12 (Additional file 1: Fig. S4B). While, as mentioned previously, cluster 3 was difficult to interpret biologically given its lack of clear expression markers (Additional file 1: Fig. S3C), clusters 4, 8, and 12 represent putative cardiomyocytes, cardiac precursors, and epithelial cells, respectively, suggesting that these cell types may have stronger determination upon seeding for differentiation. Taken together, the tendency for cells within some barcoded clones to share more similar expression states than would be seen by random chance suggests that cell type determination can occur early in or before cardiac directed differentiation, at least for some ultimate cell types.

### Survival during cardiac directed differentiation does not depend on initial hiPS cell differences

Many cells die during the process of cardiac directed differentiation, with only a fraction of initial cells giving rise to the final population of differentiated cells (see "Methods"). The substantial amount of cell death during differentiation led us to wonder, are the progeny of certain hiPS cells predetermined to survive, or does the selection process occur randomly?

To test whether hiPS cells might be predetermined to survive directed differentiation, we allowed barcoded hiPS cells to divide 3–4 times before splitting them in thirds (the number of divisions was chosen to ensure cells derived from each hiPS cell precursor were represented in each split). Two of the thirds were seeded in two separate wells to be differentiated in parallel, while the final third was harvested for genomic DNA and sequenced to determine the barcodes present in the initial population of hiPS cells— the numbers of initial barcodes recovered were roughly what we anticipated based on our estimate of the number of viruses used for barcode transduction. We sequenced the

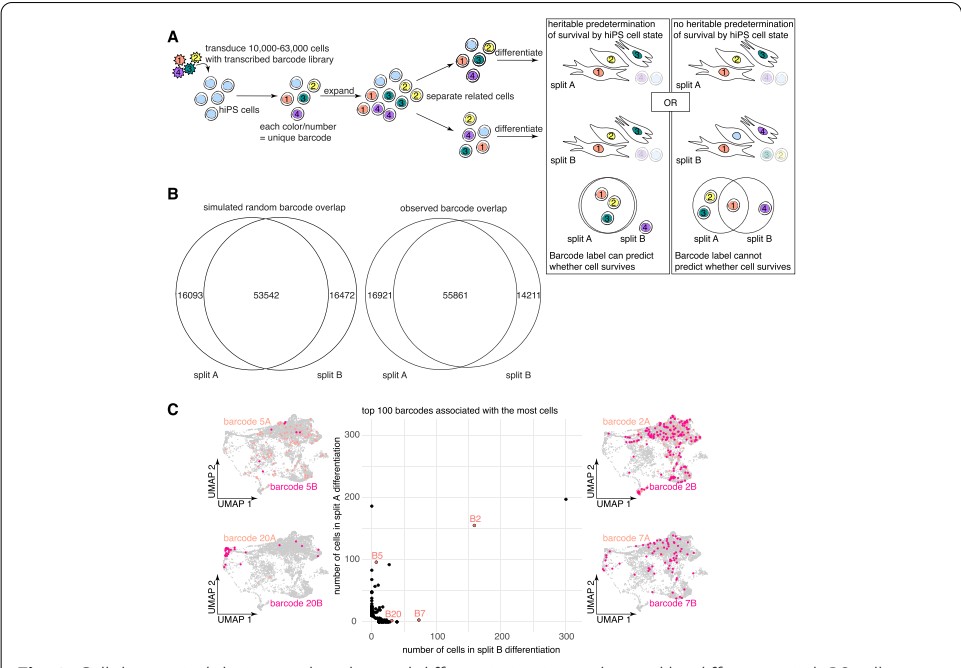

**Fig. 3** Cellular survival during cardiac directed differentiation is not dictated by differences in hiPS cells. **a** Schematic representation of experimental workflow. Briefly, we transduced hiPS cells at a range of MOIs (0.1, 0.23, 0.5, or 1.0), then 3 days later (3–4 population doublings), we harvested a third of cells for immediate extraction of genomic DNA and split the remaining cells across two parallel cardiac differentiations. On day 15 of differentiation, we harvested differentiated cells from both splits for genomic DNA extraction. We sequenced and recovered barcodes from the genomic DNA from each split, asking whether the number of overlapping barcodes between splits was greater than would be found by random chance (which would suggest that barcoded clones are predisposed to survival vs death). **b** Comparison of the simulated random barcode overlap (middle number) across splits with the observed barcode overlap from the splits differentiated from hiPS cells transduced at an MOI of ~0.5 (the experimental condition with the highest observed barcode overlap). **c** Scatter plot showing the number of cells per split labeled by the top 100 barcode clones composed of the most cells. Maintaining the organization provided by UMAP, we plotted all barcoded cells (gray) and recolored cells corresponding to each of four featured barcodes (marked in red on the scatter plot) in two shades of pink corresponding to the two parallel splits. There is little concordance for most barcode clones between the number of cells that are recovered from each split following cardiac differentiation

genomic DNA to identify the barcodes present in day 15 cells from each of the two parallel differentiations (Fig. 3A). If survival were predetermined by the intrinsic state of an hiPS cell precursor and inherited by its progeny, then the progeny of the same hiPS precursors would survive across parallel differentiations and we would recover a significant (i.e., greater than random chance) overlap in barcodes found in each split. We checked whether our data were consistent with pure chance by constructing simulations of barcode dynamics during the division and survival process of cardiac directed differentiation—we extracted and estimated the abundance of barcode sequences from the initial hiPS cell genomic DNA and used the binomial distribution to computationally "split" normalized barcode reads into two populations, modified by a loss coefficient to account for cell survival (see "Methods"). Across multiple barcoded differentiations, our observed overlap between differentiated splits was not substantially different from what we observed when simulating survival as a random binomial process (Fig. 3B, Additional file 1: Fig. S5A). Upon running this simulation 1000 times and comparing the observed

fraction overlap for each split to the simulated distribution assuming random survival, we found no meaningful differences between the two (Additional file 1: Fig. S5B), showing that our data did not necessitate the need to postulate heritable survival during differentiation.

Consistent with random survival during differentiation, even when barcodes were shared between differentiation splits they were not always present in similar abundance. In our single-cell RNA sequencing experiment described above, the two replicate differentiations were seeded from the same pool of transduced hiPS cells. In that experiment, we queried whether surviving cells across differentiations were clones derived from the same hiPS precursors and found an even starker lack of concordance in abundance of the same barcode across differentiations (Fig. 3D, Additional file 1: Fig. S5D). This lower barcode overlap may be due to subsampling as a result of the lower multiplicity of infection used for the single-cell RNA sequencing experiment and subsequent sorting for GFP-positive cells (Additional file 1: Fig. S5C). Regardless, across multiple experiments, we found that our data suggest that barcoded hiPS cells do not display intrinsic bias for survival or death during cardiac directed differentiation.

### Determination of differentiated expression states occurs after seeding of hiPS cells for cardiac directed differentiation

We wondered whether, independent from their ability to survive, hiPS cells prior to seeding for differentiation were already "primed" to become specific cell types following cardiac directed differentiation. Our earlier analysis (Fig. 2) demonstrated that differentiated cells descended from the same hiPS cell tended to have more similar expression states than less related differentiated cells, such that cell type determination must be occurring early in or before cardiac directed differentiation. Early determination could mean that cell type determination was occurring at the level of the hiPS precursor, such that intrinsic differences in hiPS precursors led to differences in differentiated expression state (i.e., priming). Alternatively, at least some cell type determination could occur shortly after seeding for differentiation, potentially driven by newly induced intrinsic differences or cell-extrinsic cues such as paracrine signaling from neighbor cells or local differences in small molecule concentration. In the intrinsic case, cell type determination has already occurred within hiPS cells such that they are already predisposed toward certain differentiated expression states when seeded for differentiation. Operationally, we can ask: if we seeded the same cells twice for differentiation, scrambling extrinsic cues like neighboring cells and position in between, would the same initial hiPS cells attain the same final expression states? This type of priming in other systems can be maintained through multiple cell divisions (i.e., is heritable [38, 47, 51, 52]), so making the assumption that it would hold for our system, we split sibling hiPS cells into two parallel differentiations. If expression states were predetermined by intrinsic differences in hiPS cell precursors, then the progeny of these separated siblings would be more transcriptionally similar to each other than to progeny of less related hiPS cells. Accordingly, to test whether these heritably primed cells exist, we transduced hiPS cells with barcode-laden lentivirus and allowed them to divide approximately three times before splitting them to be seeded for two parallel differentiations. On day 14 of differentiation, we used a GFP sort to enrich for cells from each parallel differentiation that have

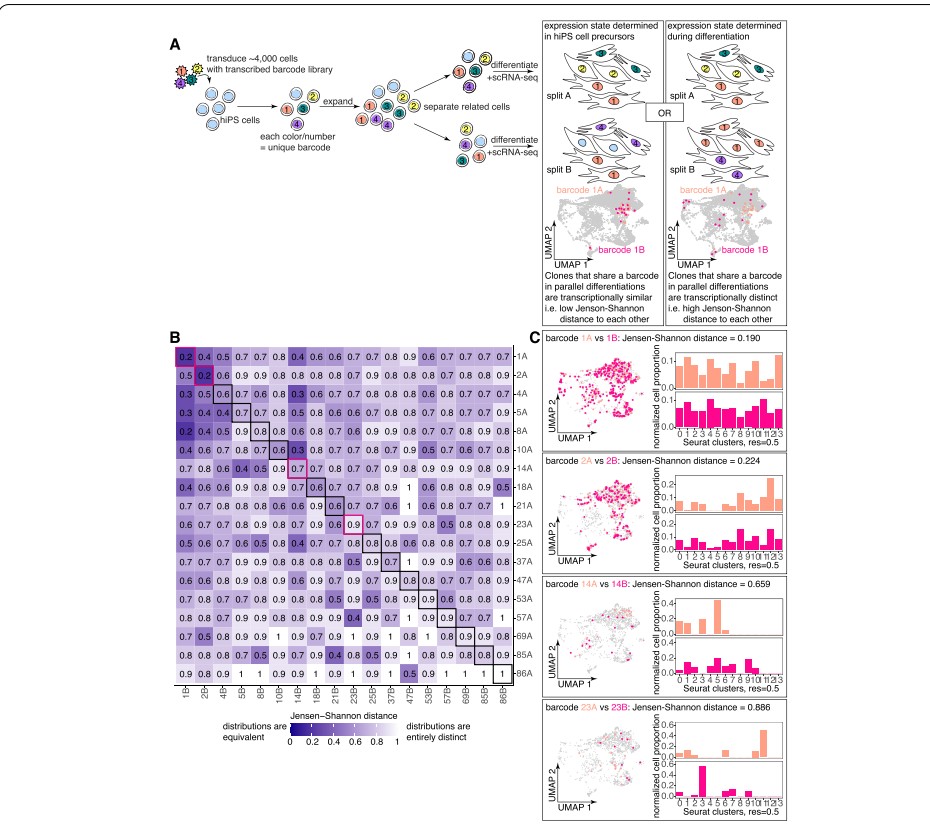

**Fig. 4** Differentiated expression state is determined after hiPS cells are seeded for differentiation. **a** Schematic representation of experimental workflow. Briefly, we transduced hiPS cells at an MOI of ~0.03, allowed them to grow for 3 days (3–4 population doublings), and split them evenly across two parallel cardiac differentiations (split A and split B). On day 14 of differentiation, we performed single-cell RNA sequencing on both splits. We asked whether barcoded clones present in both differentiations were also transcriptionally similar to get at the question of whether cell type determination occurs before or after seeding of hiPS cells for differentiation. **b** Heatmap of pairwise Jensen-Shannon distances between cells associated with 18 barcodes in split A and the cells associated with the same 18 barcodes in split B. In general, separated clones sharing a barcode (bolded outline along the diagonal) had Jensen-Shannon distances in the same range as clones labeled by distinct barcodes (off the diagonal). **c** Plots associated with the four featured barcodes outlined in pink in **b**. Maintaining the organization provided by UMAP, we plotted all cells in the analysis (gray) and recolored cells corresponding to each of the four featured barcodes in two shades of pink corresponding to the two parallel splits. Also for each featured barcode, we plot bar graphs for observed cluster probability distribution in split A (salmon) and the observed cluster probability distribution in split B (magenta)

lentiviral barcode transcripts and performed single-cell RNA sequencing on those GFP-positive cells, from which we identified clones that survived differentiation in both splits (Fig. 4A).

To address the question of whether these separated clones were more similar to each other than to a randomly selected distinct clone of cells, we once again chose to use the Jensen-Shannon distance. This time, instead of measuring the distance of an observed distribution of cells belonging to a clone across clusters from the roughly uniform random cluster probability distribution, we compared the two cluster probability distributions associated with the same barcoded clone in parallel differentiations. For the 18 barcode clones consisting of at least 5 cells in each split, we calculated the pairwise

Jensen-Shannon distances between all cluster probability distributions (Fig. 4B). The Jensen-Shannon distance values between separated clones labeled by the same barcode (on the diagonal in Fig. 4B) were in the same range as the distances between clones labeled by distinct barcodes (off the diagonal in Fig. 4B) for all but two barcodes. This was also borne out by visual inspection of the cluster probability distributions for barcoded clones, which generally appeared quite different between splits (Fig. 4C). The overall trend toward dissimilar cluster probability distributions between clones that were differentiated separately suggests that final differentiated expression state is not broadly predetermined by intrinsic differences in hiPS cell precursors, at least at this time scale. The exceptions to this trend were barcodes B1 and B2, which displayed higher within-clone similarity across split differentiations; interestingly, these barcode clones contained many more cells than the other barcode clones in our analysis. Even if we were to take the B1 and B2 results without caveats, we could only interpret them to mean that a small subset of hiPS cells may be primed to take on certain ranges of final expression states. We found a similar trend even when we analyzed clones with a minimum of 7, 10, 15, or 20 cells (Additional file 1: Fig. S6A). In contrast, when performing this analysis on barcoded vemurafenib-treated melanoma cells, a system with known heritable predetermination of the final cell state [47, 51–53], barcoded clones in one split were far more similar (i.e., had lower Jensen-Shannon distance values) to the related clones in the other split than to clones labeled by a distinct barcode (Additional file 1: Fig. S6B). An important consideration is that we analyzed many more clones with smaller numbers of cells; it is possible that the numerous small clones appear to have lower intrinsic potential due to low sampling and are masking a more obvious intrinsic potential in large clones. It may also be that the differences in intrinsic potential may reflect biological differences between clones; further experiments with more sampling would be required to distinguish these possibilities. We conclude that, in most cases, cells sharing an hiPS cell progenitor that are seeded across parallel differentiations do not take on more transcriptionally similar profiles than would be expected by random chance.

### Spatial organization of cell type markers is consistent with early, maintained cell type determination in cardiomyocyte differentiation

We did not find widespread heritable predetermination of expression state upon cardiac directed differentiation that we could trace back to the hiPS cell state. At the same time, differentiated cells that shared a barcode tended to display more clustered gene expression states than would be seen by random chance. These findings suggested that cell type determination during cardiac directed differentiation generally occurred some time after hiPS cell seeding, but it remained unclear why some barcoded clones had relatively homogeneous expression states while others displayed a wide range of expression states. Clones with strong homogeneity in expression states likely initiated cell type determination early and uniformly adopted and maintained the associated expression states throughout differentiation. However, for the clones with more variability in expression states across constituent cells, there are two potential drivers for this expression state variability. One possibility is that, among the more variable clones, cell type determination is initiated later after constituent cells have divided and diverged in expression states, such that individual cells adopt and maintain a variety of expression

states. Relatedly, a multipotent cell could, over multiple cell divisions, progressively lineage-restrict and give rise to multiple cell types with disparate cell type expression over the differentiation time course all within the same clone. Alternatively, cell type determination could be initiated early in these more variable clones but the expression state could be poorly maintained, such that over time, derived cells adopt a variety of different expression states.

To distinguish these possibilities, we performed single-molecule RNA fluorescence in situ hybridization (RNA FISH) [54] on cells in their differentiation wells for multiple cell type markers and looked to see whether cell types grew out in patches as a proxy for the maintenance of cell expression states. We hypothesized that if cell type determination initiated early in differentiation and the associated expression states were maintained through much of the cellular expansion observed in differentiation, we would observe large patches of cells of the same type (i.e., expressing high levels of the same marker). If instead cell type determination occurred late but associated expression states were maintained for the remainder of differentiation, we would expect to see small patches of cells of the same type. In contrast, if cell expression states were not maintained through the cell growth observed in differentiation, we would observe large swaths of cells consisting of interspersed cell types in the differentiation well (Fig. 5A). On day 12 of differentiation, shortly before our previous timepoints and 4 days after we first noticed contractile activity, we imaged cells to capture prevalence of contractile activity before fixing the cells in their wells (Additional files 4 and 5: Movies S1 and S2). We probed for the expression of 5 canonical cell type marker genes by single-molecule RNA FISH—*TNNT2* for cardiomyocytes, *LUM* for fibroblasts, *EPCAM* for epithelial cells, *ISL1* [19, 55] which is expressed in cardiac progenitors, and *WT1* for epicardium [56]. We stained nuclei with DAPI and then imaged multiple positions throughout each well, capturing expression of each marker gene in at least 17 images. To determine patch size, we counted the number of nuclei associated with cell patches displaying marker gene expression within each image (Fig. 5B).

All queried cell types were associated with a range of cell patch sizes, with a substantial number of patches larger than 20 cells in size (Fig. 5C). While the most common patch size was between 1 and 5 cells for all cell types, different cell types displayed differing frequencies of large patches (sometimes composed of hundreds of cells) such that the vast majority of imaged *EPCAM*-high (96%), *TNNT2*-high (86%), and *LUM*-high (86%) cells were found in patches with upwards of 20 cells. Indeed, of the profiled cell types, *EPCAM*-high cells were most commonly in very large patches (64% of imaged cells belonged to patches with upwards of 100 cells), while *WT1*-high cells tended toward smaller patch sizes (48% of cells belonged to patches with 10 cells or fewer). While we cannot comment on the precise timing of cell type determination, it seemed that once a cell type determination was made, that determination was maintained in a cell-autonomous manner for at least some divisions, resulting in patches of marker expression observable by visual inspection. A prediction from our earlier finding that cardiomyocytes, cardiac precursors, and epithelial cells might have stronger cell type determination upon seeding for differentiation (Additional file 1: Fig. S4B) is that these cell types should grow out in larger patches. We confirmed this prediction from our RNA FISH results, where we observed that *TNNT2*-high, *ISL1*-high, and *EPCAM*-high

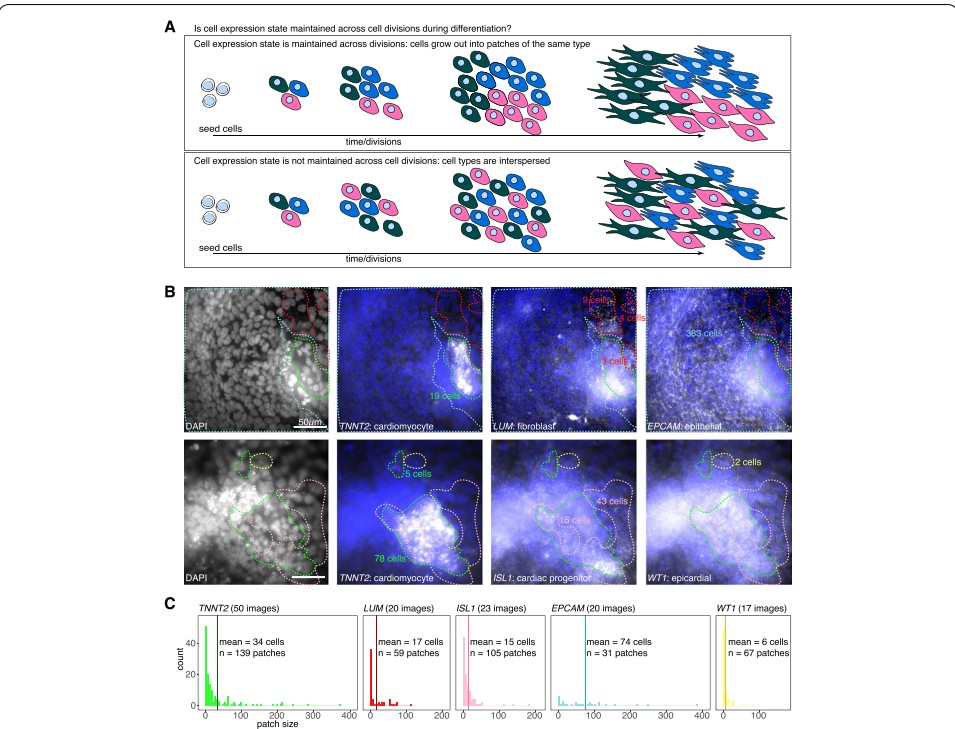

**Fig. 5** Spatial organization of cell type markers is consistent with maintenance of cell type determination in cardiomyocyte differentiation. **a** Schematic representations of how cell types might grow out during differentiation with and without maintenance of cell expression state (denoted by color). **b** Representative images of *TNNT2*, *LUM*, *EPCAM*, *ISL1*, and *WT1* single-molecule RNA FISH of cells fixed on day 12 of cardiac differentiation, demonstrating patches of cells (denoted using dotted lines—*TNNT2* in green, *LUM* in red, *ISL1* in pink, *EPCAM* in blue, and *WT1* in yellow) with high marker expression. Nuclei in patches counted manually, guided by DAPI expression. Scale bars are 50 µm. **c** Frequency of patch sizes associated with each marker gene (bars colored to match dotted lines in **b**). The mean patch size is denoted with a slightly darker colored vertical line. Total number of images and clusters analyzed per marker also indicated on each histogram

cells all were found in patches on average of 15 or more cells, whereas *WT1*-high cells were found in patches of 6 cells, on average. Therefore, these data raise the possibility that local early environmental cues may be important for the clustering of particular cell states we observed. Less likely explanations for the observed marker patterns may be that similar cell types migrated toward each other to form these patches, or that unrelated, non-dividing cells in close proximity were all induced to become similar cell types, potentially through self-organizing properties. Further work will be needed to conclusively eliminate such possibilities. Interestingly, our cell type markers displayed a range of distribution patterns, suggesting that determination may be initiated in distinct ways for different cell types. While further analysis will be necessary to definitively answer the questions proposed, our results are suggestive of a model of cardiac differentiation in which cell type determination occurs with different initiation times between cell types, following which this determination is maintained in a largely cell-autonomous manner.

## Discussion

Here we have demonstrated that cell type determination across multiple cardiac and non-cardiac cell types of interest occurs only after seeding for cardiac directed differentiation, with intrinsic differences between hiPS cell precursors playing little to no role. Nor does there appear to be an intrinsic bias among individual hiPS cells toward survival or death. Given our experimental design, where we split recently divided sibling hiPS cells across parallel differentiations, we should have been able to detect any intrinsic predetermination that was maintained through at least 3–5 cell divisions, suggesting that extrinsic factors such as location in the differentiation well or local inductive cues are likely more pertinent to final cell type determination, though we cannot exclude the possibility that intrinsic effects on shorter time scales may also be important.

While differences in the hiPS cell population likely do not predetermine final cell type following cardiac directed differentiation, the finding that, within a differentiation, cells that share an hiPS cell progenitor tend to adopt expression state trajectories more similar than random, suggests that cell type determination can occur early enough to allow for multiple cell divisions after the determination is made, at least for some cell types. Additionally, expression states characteristic of cardiomyocytes, cardiac precursors, or epithelial cells seem to be largely restricted to certain clonal populations and vice versa—barcoded clones for which a plurality of cells became one of these 3 cell types were the most internally similar, while other clones comprised a more random distribution of types. It may thus be that cardiomyocytes, cardiac precursors, and epithelial cell types have earlier initiation of determination. These results were largely recapitulated by our single-molecule RNA FISH-based analysis of how differentiated cell types are spatially distributed, where we found that all queried cell types (cardiomyocytes, fibroblasts, epithelial cells, cardiac progenitors, and epicardial cells) were present in patches that were observable by visual inspection, suggesting that cell types, once determined, are maintained in a cell-autonomous way across multiple cell divisions.

There are multiple models that might explain both the variability we observed in marker patch size by RNA FISH and the variable degree to which clonally related cells share similar final expression states. One possible interpretation of these data is that cell type determination may begin at various times during differentiation, with some determination for some cells (and cell types) occurring early in differentiation before dividing numerous times and accounting for larger cluster sizes and other cells becoming determined later after many divisions. Alternatively, there may exist differences in division rate between cells generating large and small cluster sizes despite similar timing of cell type determination, or small clusters may be generated by migrating cells. Additionally, although cell type seemed to be maintained for at least some divisions across all queried cell types, it is interesting to note that our cell type markers were found in patches with widely varying numbers of cells, suggesting that the initiation of cell type determination may vary by cell type. For instance, in a model where cell type determination is ongoing over time, one possibility is that cells have a constant probability per unit time (i.e., rate) of determining to become any of the queried cell types. The cell types associated with larger clusters (ex. epithelial) may in that case be associated with higher rates of initiation than cell types tending toward smaller clusters (ex. epicardial). Alternatively, cell types associated with smaller clusters may be less proliferative.

We propose a model in which at least some cardiac cells are determined early in differentiation. This model is supported by another recent finding that perturbations at the stage of cardiac mesoderm specification during directed differentiation led to no significant changes to the overall proportion of cardiomyocytes and non-myocytes [57], suggesting that cells may determine whether they are cardiac or non-cardiac cell types in the first few days of directed differentiation. Such a model would also be consistent with in vivo experiments that point to the existence of a cardiac multipotent cell prior to gastrulation [12–16]. Simultaneously, strict spatiotemporal patterning of *EPCAM* at the onset of murine gastrulation and in embryonic stem cell differentiation has been shown to direct the appropriate segregation of *EPCAM*-positive endoderm and *EPCAM*-negative mesoderm and is necessary for cardiomyocyte production [58]. Our finding that *EPCAM*-high cells tended more than any other queried cell type to be present in larger patches following directed differentiation and adjacent to but not co-expressed with *TNNT2* is consistent with proposed models in which *EPCAM*-high cells are also determined in the first few days of directed differentiation and facilitate cardiomyocyte maturation, potentially through physical contact or secreted factors [59–61]. In our work, we were unable to disentangle time from the number of cell divisions since cell type determination. Further work following clonal cells in space and time as they proceed through directed differentiation will be required to map the timing and additional details of cell type determination.

It is often anecdotally noted that substantial cell death occurs during the cardiac directed differentiation process [11], but heretofore it has not been known how this relates to the starting population of hiPS cells. Because efficiency in the field is often measured by percentage of differentiated cells that display markers or contractile activity, no comment is made about the extent of selection that may occur prior to this point. Here we show that, during cardiac directed differentiation only a fraction of hiPS precursor cells contribute to the final differentiated population. It could be that all cells within particular barcode clones are all fated for extermination, or alternatively that cells are probabilistically selected out from within cells across all barcode clones equally. We cannot fully distinguish between these possibilities; however, our data suggest the latter scenario is more likely. It is also possible that differences in proliferation, environmental cues, or self-organizing behavior underlie the biases in clonal behavior we observed. We note that we recovered a smaller proportion of barcodes from differentiated cells in our single-cell RNA sequencing experiments than in our genomic DNA barcode sequencing and subsequently observed a smaller proportion of overlapping barcodes between parallel differentiations. One contributor to this process may have been transgene silencing, which we found does not occur in a barcode-specific manner (Additional file 1: Fig. S5C), such that when profiling barcodes from GFP-positive cells alone we subsampled to some extent the total progeny of any hiPS progenitor cell. This subsampling was potentially compounded by our use of a lower multiplicity of infection so as to meet the technical requirements of single-cell RNA sequencing lanes; because we started with a smaller initial number of barcodes in our hiPS progenitor population, if we assume a constant low level of cell/barcode loss, our barcode recovery from differentiated cells may have been more affected than experiments where we had more initial barcodes (i.e., higher MOI).

Our results have several parallels with other reprogramming and differentiation systems that have similarly found that "elite" primed cells can be short lived or functionally absent within certain progenitor populations. Hanna et al. showed that reprogramming happens in transient populations [62], and Biddy et al. use lineage tracing to show that reprogramming outcome was different even between sibling cells [37]. It may be that some of these cell states have a very short persistence time or that microenvironmental factors are dominant.

We focused on cell type determination during cardiac directed differentiation of hiPS cells because the protocols are well-characterized, and there are distinct cell types that form following differentiation of clonal cell lines. It will be interesting in the future to apply our framework to later starting populations in cardiac directed differentiation to narrow down the timing of cell type determination. Additionally, we are increasingly able to differentiate hiPS cells to recapitulate numerous other human tissues. Application of our framework to other tissue populations may reveal universal characteristics of cell type determination during differentiation.

## Conclusions

Here we have narrowed down the timing of cell type determination and potential contributors to population heterogeneity during cardiac differentiation, showing that intrinsic differences in hiPS cells likely do not affect choice of final cell type. Future experiments will be needed to reveal when and how the events of cell type determination occur during directed differentiations toward cardiac and other tissue lineages.

## Methods

### Cell culture

#### *Generation and maintenance of hiPS cell culture*

PENN123i-SV20 hiPS cell line was generated as previously described [45], thawed, and maintained in feeder-free conditions on Geltrex (ThermoFisher, cat. A1413301)-coated dishes in StemMACS iPS-Brew-XF medium (Miltenyi, cat. 130-104-368) supplemented with 1% penicillin/streptomycin (Invitrogen, cat. 15140-122). We performed all hiPS culture incubations at 37 ℃, 5% $CO_2$, 5% $O_2$ and changed the medium every 48 h. For maintenance, we split cells every 4–5 days at a 1:10 ratio using StemMACS Passaging Solution XF (Miltenyi, cat. 130-104-688).

#### *Cardiac directed differentiation of hiPS cells*

To perform cardiac directed differentiation, we adapted standard protocols as previously described [39, 44, 49, 63]. Briefly, we grew undifferentiated hiPS cells in feeder-free conditions for 4–5 days until they reached ~75% confluency. At this point, we used a 4-min incubation with Accutase (Sigma, cat. A6964) to detach hiPS cells and subsequently counted and seeded cells onto Geltrex (Thermo Fisher, cat. A1413301)-coated 12-well plates at a density in the range of $3–7 \times 10^5$ cells per well in iPS-Brew + 2μM Thiazovivin (Sigma, cat. SML1045-5MG). Alternatively, for RNA FISH experiments, we seeded cells onto Geltrex-coated Nunc Lab-Tek 8-well chambered coverglass (Thermo Scientific, cat. 12-565-470) at a density of $4–5 \times 10^4$ cells per well. At this time (day -2), we moved cells to incubators set to 37 ℃, 5% $CO_2$, and ambient $O_2$. After 24 h, we changed

the culture medium to iPS-Brew + 1μM Chiron 99021 (Cayman Chemical, cat. 13122) and incubated cells for another 24 h. On day 0 to induce cardiac differentiation, we replaced the medium with RPMI/B27-insulin (RPMI 1640 (Invitrogen, cat. 11875085) with 2% B-27 Supplement Minus Insulin (Life Technologies, cat. 17504-044)) medium supplemented with 100 ng/mL recombinant human/mouse/rat activin A (R&D systems, cat. 338-AC- 010) and incubated cells for 18 h. On day 1, we changed the medium to RPMI/B27-insulin supplemented with 10 ng/ml BMP4 (Peprotech, cat. AF-120-05ET) + 1 μM Chiron 99021 and incubated cells for 48 h. On day 3, we changed the medium to RPMI/B27-insulin supplemented with 1 μM Xav 939 (Tocris Bioscience, cat. 3748), and incubated cells for another 48 h. During this incubation, substantial cell death was typically observed, as noted in the protocol. On day 5, we replaced the medium with RPMI/B27-insulin without supplementary cytokines for 72 h. For every 2 days after, we replaced the medium with RPMI/B27+insulin (2% B-27 Supplement Minus Insulin (Invitrogen, cat. 17504044)) + 0.5% penicillin/streptomycin + 1% L-glutamine (Invitrogen, 25030-081). We typically observed spontaneously beating activity beginning between days 8 and 12.

### Barcode lentivirus library generation

Barcode libraries were constructed as previously described [47]. Full protocol available at https://www.protocols.io/view/barcode-plasmid-library-cloning-4hggt3w. Briefly, we modified the LRG2.1T plasmid, kindly provided by Junwei Shi, by removing the U6 promoter and single guide RNA scaffold and inserting a spacer sequence flanked by EcoRV restriction sites just after the stop codon of GFP. We digested this vector backbone with EcoRV (NEB) and gel purified the resulting linearized vector. We ordered PAGE-purified ultramer oligonucleotides (IDT) containing 30 nucleotides homologous to the vector insertion site surrounding 100 nucleotides with a repeating "WSN" pattern (W = A or T, S = G or C, N = any) and used Gibson assembly followed by column purification to combine the linearized vector and barcode oligo insert. We performed 9 electroporations in total of the column-purified plasmid into Endura electrocompetent *Escherichia coli* cells (Lucigen) using a Gene Pulser Xcell (Bio-Rad), allowing for recovery before plating serial dilutions and seeding cultures (200 mL each) for maxipreparation. We incubated these cultures on a shaker at 225 rpm and 32 °C for 12–14 h, after which we pelleted cultures by centrifugation and used the EndoFree Plasmid Maxi Kit (Qiagen) to isolate plasmid according to the manufacturer's protocol, sometimes freezing pellets at −20 °C for several days before isolating plasmid. Barcode insertion was verified by polymerase chain reaction (PCR) from colonies from plated serial dilutions. We pooled the plasmids from the 9 separate cultures in equal amounts by weight before packaging into lentivirus. We estimated our library complexity as described elsewhere [52]. Briefly, we sequenced three independent transductions in WM989 A6-G3 melanoma cells and took note of the total and pairwise overlapping extracted barcodes. Using the mark-recapture analysis formula, we estimate our barcode diversity from these three transductions to be between 48.9 and 63.3 million barcodes.

### Lentivirus packaging and transduction

To package lentivirus, we adapted protocols that have been previously described [47, 64]. Prior to plasmid transfection, we grew HEK293FT cells to 80–90% confluency in $10 \times 15$-cm plates in DMEM containing 10% FBS without antibiotics. For each 15-cm plate, we added 184 μL of polyethylenimine (Polysciences, cat. 23966) to 1.15 mL of Opti-MEM (Thermo Fisher Scientific, cat. 31985062), separately combining 11.5 μg of VSVG and 17.25 μg of pPAX2 and 16.89 μg of the barcode plasmid library in 1.15 mL of Opti-MEM before incubating both solutions at room temperature for 5 min. We mixed both solutions together by vortexing and incubated the combined plasmid-polyethylenimine solution at room temperature for 15 min. We added 2.509 mL of the combined plasmid-polyethylenimine solution dropwise to each 15-cm dish. After 7 h, we aspirated the media from the cells and added fresh DMEM containing 10% FBS and 5% penicillin/streptomycin. The next morning, we aspirated the media, washed the cells with $1\times$ DPBS, and added fresh DMEM containing 10% FBS and antibiotics. Approximately 9 h later, we transferred the virus-laden media to an empty, sterile DMEM bottle and added another 10–15 mL of DMEM containing 10% FBS and antibiotics to each plate. We collected virus-laden media twice more over the next 25 h and, during this time, stored the collected media at 4 ℃. After the final collection, we filtered the virus-laden media through a 0.22-μm PES filter and then concentrated viral particles by ultracentrifugation at 24,000 rpm for 1.5 h (4 ℃) with an SW-32Ti swinging bucket rotor (Beckman Coulter). We resuspended the viral pellet in iPS-Brew at 1/200th the original supernatant volume and then stored 50–100 μl aliquots at − 80 ℃.

In preparation for transduction of hiPS cells, we grew undifferentiated cells in feeder-free conditions for 4–5 days until they reached ~75% confluency. The day prior to transduction, we used a 4-min incubation with Accutase to detach hiPS cells and subsequently counted and seeded cells onto Geltrex-coated 6-well plates at a density of $1 \times 10^5$ cells per well in iPS-Brew + 2μM Thiazovivin. The next day, to transduce hiPS cells, we added freshly thawed (on ice) virus-laden media (diluted 1:100 in iPS-Brew for all volumes < 10 μl) and polybrene (final concentration, 6 μg/mL) to the cells that had been seeded the day prior. We incubated the cells with virus for 6–8 h and then removed the media, washed the cells with $1\times$ DPBS, and added 3 mL of iPS-Brew to each well. We determined viral titers by measuring the percentage of GFP-positive cells by flow cytometry, 48 h after transduction with serial dilutions of concentrated virus. We also determined titers for virus-laden media that had been stored at 4 ℃ for 4 days following fresh thaw on ice. We used dilutions of virus that produced less than 40% GFP-positive cells to calculate the multiplicity of infection (MOI) and titer.

For single-cell RNA sequencing experiments, we used an estimated MOI of 0.04, resulting in approximately 3900–7800 initial barcodes. For genomic DNA barcode sequencing experiments, we infected cells on 2 days with estimated MOIs of 0.1, 0.23 or 0.5, and 1.0.

### Single-cell isolation and flow sorting

On the day of collection, we used 0.25% trypsin-EDTA (Gibco, cat. 25200-056) to lift cells. To quench trypsinization, we used RPMI 1640 +20% FBS following which we used pipetting to generate a single-cell suspension. After a $1\times$ DPBS (Invitrogen, cat.

14190-136) wash, we pelleted then resuspended cells in fluorescence-activated cell sorting (FACS) buffer consisting of 1% BSA + 2 mM EDTA (from 0.5M EDTA (Life Technologies, cat. 15575-020)) in 1× DPBS, supplemented with 2 μM Thiazovivin and 1× antibiotic-antimycotic (Gibco, cat. 15240-062). Cells were sorted on a MoFlo Astrios machine (Beckman Coulter). In preparation for single-cell RNA sequencing, after gating for live cells and singlets, we collected 15,000 events from each of two GFP-positive populations and 300,000 events from each of two GFP-negative populations.

### Single-cell RNA sequencing

With the day 14 sorted cells, we used the 10X Genomics single-cell RNA-seq kit v3. Briefly, we spun down and resuspended both sorted GFP-positive populations in their entirety according to the Chromium Next GEM Single Cell 3′ Reagent Kits v3.1 manufacturer directions (10X Genomics) targeting 10,000 cells for recovery. To match these numbers, we counted 15,000 cells from one of the GFP-negative populations and resuspended them in the same way. With the help of the Wistar Genomics Facility, we generated gel beads-in-emulsion (GEMs) using the 10X Chromium system (10X Genomics, Pleasanton, CA). From these GEMs, we extracted barcoded cDNA according to the post-GEM RT-cleanup instructions and amplified cDNA for 11 cycles. With 10 μl of this amplified cDNA, we proceeded with fragmentation, end-repair, poly A-tailing, adapter ligation, and 10X sample indexing per the manufacturer's protocol. We quantified libraries using the Qubit Fluorometer (Thermo Fisher) and Bioanalyzer (Agilent) analysis prior to sequencing on a NextSeq 500 machine (Illumina) using 28 cycles for read 1, 55 cycles for read 2, and 8 cycles for each index.

### Bioinformatics processing of single-cell RNA sequencing expression data

Upon downloading our NextSeq sequencing run(s), we mapped reads to the original transcripts and cells using the cellranger pipeline v4.0.0 by 10X Genomics. We began by using *cellranger mkfastq* with default parameters to demultiplex raw base call files into library-specific FASTQ files. To align FASTQ files to the GRCh38p13 human reference genome and extract gene expression counts matrices, we used *cellranger count* based on Gencode v32 annotation, filtering and correcting cell identifiers and unique molecular identifiers (UMI) with default settings. Prior to filtering, we recovered 6117 and 4856 estimated differentiated cells for each of our GFP-positive samples, respectively, and 6645 estimated differentiated cells for our GFP-negative samples. From here on out, most of our single-cell expression analysis was done in Seurat v4 [46]. Within each experimental sample, we removed genes that were present in less than 3 cells and cells with less than or equal to 200 or greater than or equal to 10,000 genes. Post filtering, we performed normalization and variance stabilization within each sample using Seurat's *SCTransform* algorithm [65]. We integrated all differentiated samples according to the Satija lab's integration workflow (https://satijalab.org/seurat/articles/integration_introduction.html), anchoring to the 5000 most variable genes. We used this integrated dataset to generate data dimensionality reductions by principal component analysis (PCA) and Uniform Manifold Approximation and Projection (UMAP), using 50 principal components for UMAP generation. We also tested integration using 1000, 2000, and 7000 variable genes. Although there were small differences between dimensional reductions

generated from integrated datasets anchored to 1000 vs 2000 vs 5000 genes (somewhat expected, per Satija lab instructions to use 3000 or more feature anchors), we noticed essentially no differences in the dimensional reductions resulting from integration with 5000 and 7000 features, leading us to select the former.

To generate Seurat clusters for further analysis, we tested a range of resolutions with Seurat's *FindClusters* command between 0.4 and 1.2 (recommended for datasets of around 3000 cells) and examined the resulting clusters as visualized on UMAP. We chose to focus on clusters generated using a resolution of 0.5 as these best captured a putative cardiomyocyte cluster (based on high TNNT2 and ACTC1 expression) as well as grouped together dividing cells (high expression of MKI67). The processed data following filtering, normalization, and clustering was used as the input to select marker genes for each cluster as compared to all other cells using *FindAllMarkers* with the parameters "only.pos = TRUE, min.pct = 0.25" to select for only genes with positive expression in at least 25% of the cells within a cluster. We selected 2–3 of these markers per cluster to generate a heatmap of normalized RNA expression using the RNA@data slot of our Seurat object and the *ComplexHeatmap* package [66].

### Principal component analysis of differentiated and iPS single-cell RNA sequencing data sets

We sorted hiPS cells and sequenced, filtered, and normalized the resulting dataset in Seurat as described for the differentiated cells above, resulting in 1198 cells for analysis. As before, we used *SCTransform* to normalize and variance-stabilize the dataset and then performed principal component analysis using Seurat's *RunPCA* command, using the 5000 variable features used to normalize the object. To get total variance for each dataset (i.e., both the hiPS cell dataset and the integrated differentiated cell dataset), we took the sum of the variance estimates per row of the SCT@scale.data matrix (where each row represented a gene). We calculated the eigenvalues by squaring the standard deviations per principal component stored by Seurat following PCA generation. To calculate the fraction of variance explained per principal component, we divided each of our eigenvalues by the total variance, using *ggplot2* to plot the fraction of variance explained for each of the first 10 principal components. To estimate how much variance could be explained by pure chance, we also ran PCA on randomized data.

### Barcode recovery from single-cell RNA sequencing data

To extract barcode information from our GFP-positive cells, we used another 10 µl of the amplified cDNA. With this cDNA, we ran an extra PCR "side reaction" using, on one side, a range of primers that target the 3′ UTR of GFP (Additional file 2: Table S1) and on the other side a primer that targets a region introduced through the library preparation called "Read 1" (18 cycles using NEBNext Q5 Hot Start HiFi PCR Master Mix (New England Biolabs)). These primers amplify a region that contains both the 10X cell-identifying sequence as well as the lentivirally introduced 100 bp barcode, enabling us to connect barcode clone information to the expression information indexed to each 10X cell-identifying sequence. Following this, we performed a 0.7× bead purification (Beckman Coulter SPRIselect) before pooling final libraries at equimolar ratios for sequencing

on a NextSeq 500 machine using 26 cycles for read 1, 124 cycles for read 2, and 8 cycles for each index.

**Bioinformatics processing of barcoded single-cell data**

We recovered barcodes from our side reaction sequencing data using custom shell and python scripts available on GitHub [67] and Dropbox at the link below in the directory "10XbarcodepipelineScripts". These scripts search through each read searching for sequences complementary to the side reaction library preparation primers, filtering out reads that lack the GFP barcode sequence, have too many repeated nucleotides, or do not meet a phred score cutoff. To merge highly similar barcode sequences (that may have diverged due to sequencing and/or PCR errors), we took the first 30 base pairs of each barcode sequence and used STARCODE software [68], available at https://github.com/gui11aume/starcode, to merge sequences with Levenshtein distance $\leq 6$, summing the counts and keeping only the most abundant barcode sequence.

To assign barcodes to individual cells, we performed a series of filtering steps. We first filtered out all barcodes that were associated with fewer than 2 unique molecular identifiers (UMI). We expected no more than 1 barcode per cell given the low MOI used to transduce hiPS cells, but the depth of sequencing required to recover maximal barcode information can lead to spurious assignment of multiple barcodes per cell. We selected only the barcodes associated with $\geq 30\%$ of the total number of UMIs assigned to each individual cell, in many cases resolving the multiple barcodes issue by leaving only a single dominant barcode. We kept only unique cell-barcode pairs, filtering out any cells that were still assigned multiple barcodes. Altogether, we were able to recover barcode information for 5860 of the 10,973 GFP-positive cells (~53%). These 5860 cells shared 1024 unique barcodes, substantially fewer than the estimated 3900–7800 initial barcodes.

**Derivation of Seurat cluster probability distributions and Jensen-Shannon distance analysis**

For within differentiation barcode clonal analysis, we looked at barcodes labeling at least 20 cells following cardiac directed differentiation within either of the 2 differentiations. We found that only two cells labeled by these barcodes were found in Seurat cluster 14 and so removed this cluster from our analysis, focusing on clusters 0 through 13. This resulted in separate datasets of 1576 total cells associated with 30 barcodes and 950 total cells associated with 19 barcodes. For each barcode clone within a dataset, we found how its associated cells partitioned across Seurat clusters 0 through 13. We then divided the raw number of cells per cluster by the total number of cells found in that cluster within that dataset (i.e., how many of the 1576 or 950 total cells partitioned into that Seurat cluster), then normalized all cluster proportions to sum to 1 to get "probability distributions" for each barcoded clone.

To generate the probability distributions we might expect from random chance, we averaged the results from the normalization as above of 1000 random samples of a matched number of cells from the total 1576 or 950 cells in the dataset, noting that for all barcodes this distribution was approximately uniform. We calculated the Jensen-Shannon distance between our observed barcode probability distribution and the averaged random probability distribution largely as previously described [69, 70], the one

exception being that we chose to use log base 2 to calculate the Kullback-Leibler divergence, such that maximally different samples would have a Jensen-Shannon distance of 1. To determine the significance of our calculated Jensen-Shannon distance, we took another 1000 random samples of a matched number of cells, and for the probability distribution associated with each random sample, calculated the Jensen-Shannon distance from the averaged random probability distribution.

For analyses of barcode clone distribution across differentiations, we filtered for barcode clones labeling at least 5 cells in each split following parallel cardiac directed differentiations, resulting in 1450 total cells belonging to 18 sets of barcoded clones. None of these cells were found in Seurat cluster 14, so it was once again omitted from our analysis. For each barcode, we generated separate probability distributions as above for the cells found in each split, normalizing per cluster by the number of cells out of the 1450 in the analysis found in that cluster. We calculated the Jensen-Shannon distances as above between each barcode probability distribution and the 18 probability distributions associated with the other split and visualized these distances in a heatmap, where the Jensen-Shannon distances associated with the same barcode across parallel splits are found on the diagonal.

### Barcode library preparation from genomic DNA and subsequent sequencing

We prepared barcode libraries from genomic DNA as previously described [47]. Briefly, we isolated genomic DNA from cells, a subset of which had barcodes, using the QIAmp DNA Mini Kit (Qiagen, cat. 51304) per the manufacturer's protocol. Extracted DNA was stored at $-20\,°C$ for days to weeks for some of the samples before the next step. We then performed targeted amplification of the barcode vector using custom primers containing Illumina adaptor sequences, unique sample indices, variable-length staggered bases, and an "UMI" consisting of 6 random nucleotides (NHNNNN). Although these "UMIs" are not true unique molecular identifiers, we found that, as previously described, they appeared to modestly normalize read counts and increase reproducibility. To reduce PCR amplification bias, we determined the number of cycles for each cell type by first performing a separate quantitative PCR (qPCR) and selecting the number of cycles needed to achieve one-third of the maximum fluorescence intensity for serial dilutions of genomic DNA. We then performed multiple PCR reactions using the remaining total isolated genomic DNA utilizing this cycle information, followed immediately by a $0.7\times$ bead purification (Beckman Coulter Ampure XP). We pooled purified libraries, quantified them using the Qubit dsDNA High Sensitivity Assay (Thermo Fisher Scientific), and sequenced them on a NextSeq 500 using 150 cycles for read 1 and 8 cycles for each index.

To check for silencing of transgenes, prior to isolating genomic DNA from one of our conditions (MOI 1.0), we sorted out GFP-positive and GFP-negative cells from both sibling differentiation wells, preparing cells as described above. We sorted cells into $1\times$ DPBS $+$ 2 μM Thiazovivin, collecting 439,000 GFP-positive and 953,000 GFP-negative cells from one differentiation well and 737,000 GFP-positive and 1,500,000 GFP-negative cells from the other. Barcodes were found in both GFP-positive and GFP-negative sorted conditions, suggesting that transgene silencing does not occur in a barcoded clone (i.e., integration site)-specific manner. Because our GFP-positive and GFP-negative cell

populations have indistinguishable expression profiles by single-cell RNA sequencing (described above), we concluded that there likely were not major differences between barcoded cells that continued to transcribe their barcodes and those that did not, such that our analyses of barcoded cell distributions were likely representative of the barcoded population as a whole.

### Computational analyses of barcode genomic DNA sequencing data

We recovered barcodes from sequencing of our genomic DNA barcode libraries, with some adaptations to our pipeline as previously described [47]. Briefly, we used custom Python scripts available at https://github.com/arjunrajlaboratory/timemachine to search for barcode sequences that pass a minimum length and phred score cutoff. Along with counting total reads for each barcode, we also count the number of "UMIs" incorporated into the library preparation primer sequences (see above section). As before, we do not believe that these "UMIs" tag unique barcode DNA molecules, but empirically they slightly improve correlation in barcode abundance among replicate libraries. We took the first 30 base pairs of each STARCODE [68] software to merge sequences with Levenshtein distance $\leq 8$, summing counts across merged sequences and keeping the most abundant barcode sequence. We further filtered out all barcodes containing three unknown bases ("NNN") in a row.

To approximate the number of barcodes that would overlap between splits by random chance, we extracted initial barcode sequences and estimated their initial abundance by "UMI" number per barcode, where these "UMIs" serve as a potentially more accurate proxy for initial cell numbers than read count. We then used the binomial distribution to computationally "split" these initial barcode "UMIs" into two populations, each then modified by a loss coefficient to account for cells that were lost during differentiation and experimental manipulation. We chose a loss coefficient for which the simulation resulted in overlap values between initial barcode pool and barcodes recovered from either differentiated split matching those found experimentally. Between 22 and 66% of initial barcodes were also found in differentiated samples (i.e., "survival proportion")—although highly variable across experiments, the survival proportion tended to vary less between the 2 splits associated with each initial barcoded sample. Using the loss coefficient, we determined how much overlap between simulated splits might be expected due to chance alone.

### Single-molecule RNA FISH

*TNNT2* and *LUM* probes were used as previously described [49, 71]. For our other genes of interest, we designed complementary oligonucleotide probe sets using custom probe design software written in MATLAB as previously described [47, 72, 73] and ordered oligonucleotides with a primary amine group on the 3′ end from Biosearch Technologies (see Additional file 3: Table S2 for probe sequences). We pooled all complementary oligonucleotides for each gene (14–32 oligonucleotides, depending on gene length and favorable binding regions) to make a probe set, coupling the resulting probe set to Cy3 (GE Healthcare), Alexa Fluor 594 (Life Technologies), or Atto647N (ATTO-TEC) *N*-hydroxysuccinimide ester dyes. We performed

single-molecule RNA FISH as previously described [54]. To fix cells, we aspirated media from differentiated cells, washed the cells once with 1× DPBS, and then incubated the cells in fixation buffer (3.7% formaldehyde in 1× DPBS) for 10 min at room temperature. We then aspirated the fixation buffer, washed samples twice with 1× DPBS, and added 70% ethanol before storing samples at 4 °C. For hybridization of RNA FISH probes, we rinsed samples with wash buffer (10% formamide in 2× SSC) before adding hybridization buffer (10% formamide and 10% dextran sulfate in 2× SSC) with standard concentrations of RNA FISH probes and incubating samples overnight with coverslips, in humidified containers at 37 °C. The next morning, we performed two 30-min washes at 37 °C with wash buffer, after which we added 2× SSC with 50 ng/mL of DAPI. We mounted the sample for imaging in 2× SSC.

### Imaging

We imaged RNA FISH samples on a Nikon TI-E inverted fluorescence microscope equipped with a SOLA SE U-nIR light engine (Lumencor), a Hamamatsu ORCA-Flash 4.0 V3 sCMOS camera, and × 4 Plan-Fluor DL 4XF (Nikon MRH20041/MRH20045), × 20 Plan-Apo λ (Nikon MRD00205), and × 60 Plan-Apo λ (MRD01605) objectives. As multiple layers of cells are generated through cardiac directed differentiation, we acquired *z*-stacks (0.5 μm spacing between slices) encompassing multiple complete cells at × 60 magnification. To acquire different fluorescence channels, we used the following filter sets: 31000v2 (Chroma) for DAPI, 41028 (Chroma) for Atto 488, SP102v1 (Chroma) for Cy3, and 17 SP104v2 (Chroma) for Atto 647N, and a custom filter set for Alexa 594. We tuned the exposure times depending on the dyes used: 500 ms for probes in Cy3, Atto 647N, and Alexa 594 and 10 ms for DAPI probes. We also acquired images in the Atto 488 channel with a 200 ms exposure as a marker of autofluorescence.

We used the same microscope setup to take × 4 magnification brightfield short videos demonstrating the prevalence of contractile activity prior to fixation of cells for RNA FISH.

### Image processing

For each × 60 image stack, we identified by eye the extent of RNA FISH signal and counted the number of DAPI-stained nuclei corresponding to each patch (i.e., adjacent cells expressing the same marker gene) of signal. We systematically biased ourselves against large cluster sizes in the following ways: (1) we included patches of cells that clearly continued outside the field of view of the image, (2) we did not count any DAPI stains as nuclei that were atypically small or in focus out of plane with RNA FISH signal), and (3) we deemed patches as separate if there appeared to be a clear bridge smaller than 3 cell widths or if we found even one cell that appeared to not express the marker gene between other patches. We kept separate counts for each marker gene such that a cell co-expressing two markers would be counted for both markers.

## Supplementary Information

---

**Additional file 1: Figures S1-6**.

**Additional file 2: Table S1**: Primers used to recover barcodes from 10X RNA sequencing libraries.

**Additional file 3: Table S2**: Sequences of the single molecule RNA FISH probes used in this study.

**Additional file 4: Movie S1**: Contracting cardiomyocytes in wells used for single molecule RNA FISH marker studies.

**Additional file 5: Movie S2**: Contracting cardiomyocytes in wells used for single molecule RNA FISH marker studies.

**Additional file 6.** Review history.

---

### Acknowledgements

We thank members of the Raj lab, Jain lab, and Penn iPSC core, particularly Philip Burnham, Lee Richman, Lauren Beck, Samuel Reffsin, and Margaret Dunagin for insightful discussions related to this work. We thank Nancy Zhang for providing valuable advice regarding the latent dimension analysis. We thank the Genomics Facility at the Wistar Institute, especially Sonali Majumdar and Sandy Widura for assistance with sequencing and single-cell partitioning and addition of 10X cell identifiers. We thank the Flow Cytometry Core Laboratory at the Children's Hospital of Philadelphia Research Institute for assistance with flow cytometry and fluorescence-activated cell sorting. We thank Hao Wu and Peng Hu for advice regarding cardiac directed differentiation. We thank Kenneth Zaret, Naomi Takenaka, Roberto Bonasio, and Timothy Christopher for assistance with lentivirus ultracentrifugation and sequencing. We thank Jennifer Phillips-Cremins and members of her lab, especially Linda Zhou and Zoltan Simandi for advice on hiPS cell culture and for assistance with large-scale plasmid preps.

### Review history

The review history is available as Additional file 6.

### Peer review information

### Authors' contributions

CLJ, RJ, and AR conceived and designed the project with valuable input from YG, NJ, RET, and WY. CLJ designed, performed, and analyzed all experiments, supervised by RJ and AR. QW and RJ contributed to some hiPS cell maintenance and differentiation experiments. QW and RET provided guidance regarding cardiac directed differentiation protocols. YG, NJ, AJC, IAM, and KK assisted with analysis pipeline generation. Analysis pipeline for barcode retrieval and matching with single-cell RNA sequencing data was developed by YG. Protocols for retrieval of barcodes from single-cell RNA sequencing data were developed by YG, BE, and KK. The protocol for barcode library generation and subsequent initial pipeline for retrieval of barcodes from genomic DNA sequencing were developed by BE. CLJ, RJ, and AR wrote the manuscript. The author(s) read and approved the final manuscript.

### Authors' information

Twitter handle: @cojiberries (Connie L. Jiang), @yogeshgoyallab (Yogesh Goyal), @uffdaALLberries (Naveen Jain), @RTruitt_415 (Rachel E. Truitt), @allycote (Allison J. Coté), @ian_mellis (Ian A. Mellis), @MrKarunhands (Karun Kiani), @arjunrajlab (Arjun Raj).

### Funding

CLJ acknowledges support from NIH training grants F30 HG010822, T32 DK007780, and T32 GM007170; YG acknowledges support from the Burroughs Wellcome Fund Career Awards at the Scientific Interface; NJ acknowledges support from NIH F30 HD103378; AJC acknowledges support from NIH T32 GM07229; BE acknowledges support from NIH F30 CA236129, NIH T32 GM007170, and NIH T32 HG000046; IAM acknowledges support from NIH F30 NS100595; KK acknowledges support from NIH T32 GM008216; WY acknowledges support from NIH Director's Transformative Research Award R01 GM137425, NIH UG3 DK122644, NIH UM1 DK126194, DoD CDMRP-PRORP, NIH U01 TR001810, and the Perelman School of Medicine, University of Pennsylvania; RJ acknowledges support from NIH Director's Transformative Research Award R01 GM137425, Burroughs Wellcome Foundation, Allen Foundation, American Heart Association; and AR acknowledges support from NIH Director's Transformative Research Award R01 GM137425, NIH R01 CA238237, NIH R01 CA232256, NIH P30 CA016520, NIH SPORE P50 CA174523, NIH U01 CA227550, NIH 4DN U01 DK127405, NIH Center for Photogenomics RM1 HG007743, NSF EFRI EFMA19-33400.

### Availability of data and materials

All single-cell RNA sequencing data generated for this study are available on GEO (Accession GSE198729) [74]. For all scripts required to process raw data to the graphs and images included in this paper, please see: https://doi.org/10.5281/zenodo.5942547 [67]. Genomic DNA barcode sequencing data are available at https://doi.org/10.6084/m9.figshare.19125935 [75]. Barcode data recovered from single-cell RNA sequencing side reaction are available at https:/doi.org/10.6084/m9.figshare.19126985 [76]. Raw FISH images can be found on BioStudies (Accession S-BIAD319) [77]. Data can additionally be found at https://www.dropbox.com/sh/pcihymkterrvvz3/AACTEQWG2KKQw3bi0NqShsWYa?dl=0. All analyses were done in R. We used a selection of color-blind friendly colors adapted from http://mkweb.bcgsc.ca/biovis2012/color-blindness-palette.png.

## Declarations

### Ethics approval and consent to participate
Not applicable.

### Competing interests
AR receives royalties related to Stellaris RNA FISH probes. All other authors declare that they have no competing interests.

### Author details
[1]Genetics and Epigenetics, Cell and Molecular Biology Graduate Group, Perelman School of Medicine, University of Pennsylvania, Philadelphia, PA, USA. [2]Department of Cell and Developmental Biology, Feinberg School of Medicine, Northwestern University, Chicago, IL, USA. [3]Center for Synthetic Biology, Northwestern University, Evanston, IL, USA. [4]Department of Bioengineering, School of Engineering and Applied Sciences, University of Pennsylvania, Philadelphia, PA, USA. [5]Department of Medicine, University of Pennsylvania, Philadelphia, PA, USA. [6]Department of Cell and Developmental Biology, University of Pennsylvania, Philadelphia, PA, USA. [7]Institute for Regenerative Medicine, University of Pennsylvania, Philadelphia, PA, USA. [8]Cell Biology, Physiology, and Metabolism, Cell and Molecular Biology Graduate Group, Perelman School of Medicine, University of Pennsylvania, Philadelphia, PA, USA. [9]Genomics and Computational Biology Graduate Group, Perelman School of Medicine, University of Pennsylvania, Philadelphia, PA, USA. [10]Penn Cardiovascular Institute, Perelman School of Medicine, University of Pennsylvania, Philadelphia, PA, USA. [11]Department of Genetics, Perelman School of Medicine, University of Pennsylvania, Philadelphia, PA, USA.

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

## 
