## [**Additional file 6.** Review history. · Genome Biology]

Review History

First round of review

Reviewer 1

Are you able to assess all statistics in the manuscript, including the appropriateness of statistical tests used? Yes, and I have assessed the statistics in my report.

Comments to author:

Summary

In this study Jiang et al use single cell sequencing and cell barcoding methods to study the role of genetic bias during in vitro stem cell differentiation. Overall, the study uses innovative methods in barcoding and single cell technologies to dissect the biology of stem cell differentiation. Understanding the nature in which iPSCs are biased toward specific cell fates is a very important question and remains a challenge for the field. Therefore, the fundamental concept of the study is important and worth addressing. The methods appear well described and I had no issues with the wet or dry lab approaches as outlined in the methods. However, overall, the results largely indicate that differentiation is a stochastic process driven by spatial and environmental factors to instruct differentiation. Overall, the conclusions are reasonable but do not substantially alter our current understanding of the dynamics of stem cell differentiation.

Major concerns

Throughout the study, barcode single cell rna-seq was mapped to UMAP plots derived from single cell analysis in Figure 1. This is problematic because the cell diversity of any subsequent differentiation from the barcodes is likely to be different from the original differentiation because of normal variation in differentiation efficiency. Therefore, mapping across data sets creates problems with interpretation because there may be intrinsic cell type bias in the differentiation that is influencing the barcode differentiation data to create the illusion of barcode bias. I would have performed scRNA-seq on the total population of the barcode samples then map the variation of the barcoded cells against the reference point of the total population. Also, the calculation of bias is dependent on how many cells have a barcode relative to the cluster size (e.g. few barcode hitting larger clusters would give a higher probability of barcode bias). Overall, the authors make conclusions from data in figure 2 which I don't think are reasonable. Clonal bias must be justified by more substantial characterisation of the barcode population to indicate that cells had equal probability of deriving any of the diverse cell types indicated.

Furthermore, the data in Figure 2 can be explained by the fact that cells are clonally contributing to specific cell types early in differentiation because of environmental bias and proliferative properties - certain cell types are derived by self-organising and local environmental self-reinforcing gene programs that could account for any clonal bias observed at the end of differentiation. I do not think there is sufficient evidence to suggest any clonal bias based on these barcoding studies.

Figure 3-4 indicates that survival and starting gene expression state do not influence the contribution of iPSCs to differentiation potential. These experiments seem reasonably designed and interesting but the results are not surprising and do not significantly inform how we approach protocol development for differentiation.

For Figure 5 - the authors postulate that if cells displayed clonal division to give rise to restricted cell lineages then they would appear in clusters spatially in the well. I don't think this general concept is properly tested in this analysis - but instead it reflects how cells organise in a dish to differentiate - cell/cell interactions drive differentiation decisions. Evidence of clonal bias would only reflect the natural process of spatial bias in cell/cell environmental influences driving differentiation, not something intrinsic to the cells genetic predisposition. The authors say that "Less likely explanations for the observed marker patterns may be that similar cell types migrated towards each other to form these patches, or that unrelated, non-dividing cells in close proximity were all induced to become similar cell types". I disagree - I think there are natural cell micro-environmental factors that determine cell decisions and enable positional bias (clustering) in cell differentiation decisions. Therefore clonal bias is only a random consequence of what cues are received by a cell during differentiation and are completely independent of a cell autonomous differentiation bias. The data do not disprove this possibility and therefore the main conclusions of this data do not seem justified.

Reviewer 2

Are you able to assess all statistics in the manuscript, including the appropriateness of statistical tests used? Yes, and I have assessed the statistics in my report.

Comments to author:

Here, Jiang et al., use elegant single cell lineage tracing methods to investigate cardiac directed differentiation of human induced pluripotent stem cells. Most directed differentiation methods lack efficiency and fidelity. However, the mechanisms underlying these current limitations have not been fully characterized. Thus, this study represents an important contribution to the stem cell field and an excellent demonstration of the value of genomic lineage tracing to a broader cell biology discipline. By associating differentiated cells with their early ancestors, the authors were able to demonstrate that differentiated cells that share an hiPS cell progenitor were more transcriptionally similar to each other than to other cells in the differentiated population. However, this transcriptional similarity was not observed when the same hiPS cells were differentiated in parallel. Together, with analysis of the spatial distribution of differentiating cells, these results suggest that cell fate decisions within this system are made shortly after cells are seeded. Overall, this is a sophisticated study that is performed with rigor. I have the following specific comments:

Major comments:

1. Overall, I think the analysis and statistical methods used are extremely elegant. I like the selection of the the Jensen-Shannon distance as a metric of similarity. I also think the experimental design - where two-thirds of the cells are differentiated in parallel, and one-third was DNA sequences determine the barcodes present in the initial population of hiPS cells to account for cell loss - is neat. However, one potentially weaker area of the analysis lies in

whether it is sufficiently powered in some cases. For example, the authors note that 'cluster 3's behavior is likely artifactual and driven by its small size'. Can they provide some kind of formal test to support this? Furthermore, for some of the analyses, the selected clones consisted of only 5 cells in each split, which I think might be too few cells. Indeed, barcodes B1 and B2, displayed higher within-clone similarity across split differentiations, which might support the smaller clones generating a skewed outcome. We've run into similar issues with our own clonal analyses.

2. It would be helpful to include additional metrics, such as the percentage of cells captured by flow cytometry. Additionally, the authors claim that the GFP negative cell population was transcriptionally indistinguishable - it would be helpful to include these results.

3. Overall, there are several parallels between this study and reports in the cell reprogramming arena. For example, work from Rudy Jaenisch's lab rules out the existence of 'elite' cell populations that are primed to reprogram and instead suggests that cells transition through 'privileged' states in which they are more likely to reprogram (Hanna et al., Nature, 2009). Similarly, in our own lineage tracing of reprogramming, we showed that sisters within the same biological replicate shared the same reprogramming outcomes, established early in the fate conversion process. However, sisters split across parallel replicates did not share reprogramming outcomes (Bidy et al., Nature, 2009). However, the authors' experimental design in this current study to incorporate DNA sequencing to account for absent sisters is more elegant. Together, I think these studies point to the importance of studying early events in differentiation/reprogramming and the power of lineage tracing tools to enable these investigations.

Minor comments:

1. Although the barcode library is presumably highly complex, it would be helpful for the authors to include details of this in the methods and the expected homoplasmy given the sizes of the populations labeled.

2. With respect to the visualization of clones, light pink on grey is difficult to distinguish. Perhaps a different symbol could be used, maybe an x?

Reviewer reports:

Reviewer #1: Summary

In this study Jiang et al use single cell sequencing and cell barcoding methods to study the role of genetic bias during in vitro stem cell differentiation. Overall, the study uses innovative methods in barcoding and single cell technologies to dissect the biology of stem cell differentiation. Understanding the nature in which iPSCs are biased toward specific cell fates is a very important question and remains a challenge for the field. Therefore, the fundamental concept of the study is important and worth addressing. The methods appear well described and I had no issues with the wet or dry lab approaches as outlined in the methods. However, overall, the results largely indicate that differentiation is a stochastic process driven by spatial and environmental factors to instruct differentiation. Overall, the conclusions are reasonable but do not substantially alter our current understanding of the dynamics of stem cell differentiation.

We thank the reviewer for their enthusiasm for our work.

Major concerns

Throughout the study, barcode single cell rna-seq was mapped to UMAP plots derived from single cell analysis in Figure 1. This is problematic because the cell diversity of any subsequent differentiation from the barcodes is likely to be different from the original differentiation because of normal variation in differentiation efficiency. Therefore, mapping across data sets creates problems with interpretation because there may be intrinsic cell type bias in the differentiation that is influencing the barcode differentiation data to create the illusion of barcode bias. I would have performed scRNA-seq on the total population of the barcode samples then map the variation of the barcoded cells against the reference point of the total population. Also, the calculation of bias is dependent on how many cells have a barcode relative to the cluster size (e.g. few barcode hitting larger clusters would give a higher probability of barcode bias). Overall, the authors make conclusions from data in figure 2 which I don't think are reasonable. Clonal bias must be justified by more substantial characterisation of the barcode population to indicate that cells had equal probability of deriving any of the diverse cell types indicated.

We thank the reviewer for raising the critical point that single cell RNA-seq visualization and barcode mapping must be performed within rather than between differentiation datasets to avoid misinterpreting intrinsic cell type bias as barcode bias. Indeed, as the reviewer astutely suggested, we performed scRNA-seq on both the barcoded cells featured in Figures 2-4 as well as additional non-barcoded cells from the same differentiation. (We combined these sets of cells to generate the single cell analysis in Figure 1.) What we found is that there was generally not a lot of bias in terms of which barcodes went to which transcriptional clusters, although we did notice an underrepresentation bias in clusters 1, 3, and 5 (and others to a lesser degree). In each of those cases, we were unable to find transcriptional markers with high specificity for these clusters, thus making it difficult to interpret those clusters biologically anyway. For these reasons, we did not focus any of our conclusions on barcodes that came from those strongly underrepresented clusters. Overall, we do not think the bias in the other clusters was significant enough to affect our conclusions of transcriptional constraint, a point which we have now clarified in the text:

“...As transcripts of the GFP transgene are captured by single cell RNA sequencing, we were able to recover and connect Rewind barcodes to individual cells through analysis of GFP transcript 10x

sequencing reads (Supplementary Figures 2-3). We also profiled some of our GFP negative cell population and found the populations to be very similar to the GFP positive populations (Supplementary Figure 2), suggesting that the introduction of the barcode itself did not introduce major bias (there was some bias in clusters 1 and 3, but those clusters were difficult to interpret biologically due to a lack of good markers and hence were not subjected to further analysis (Supplementary Figure 3C). Through this process, we were able to both profile the expression state of individual differentiated cells and determine which differentiated cells were descended from the same barcoded hiPS cell progenitor. ...”

The reviewer additionally suggested that barcode clone bias toward specific clusters may perhaps be driven by higher proportions of barcodes within smaller clusters and asked for a more substantial characterization of the barcode population’s segregation between indicated cell types. To this end, we looked at both the number and proportion of barcoded and unbarcoded cells within each Seurat cluster in our dataset. We did not find that larger clusters consistently had smaller proportions of barcodes— instead, to the contrary, we found that aside from cluster 14 (which we had previously removed from our analyses due to its small contributions of both barcoded and unbarcoded cells), generally the smaller clusters (clusters 6-13) were actually more proportionally sampled than the bigger clusters (0-5). Thus, we were less concerned about the potential for a few barcodes hitting a larger cluster yielding a lot of bias. We thank the reviewer for bringing up these important points. We have included these findings in a modified Supplementary Figure 3 as follows:

Supplementary Figure 3

Supplementary figure 3. Barcode library complexity and cell distribution across Seurat clusters. (A) Overlap in barcodes recovered from 3 independent transductions as visualized using a Venn diagram, adapted from Goyal *et al.* 2021. Using these overlaps we estimate that we have over 45 million unique barcodes in our library (see Methods for details). (B) Maintaining the organization provided by UMAP, we recolored all GFP positive sorted cells in each differentiated split (A or B) profiled by 10X single-cell RNA sequencing (grey) and then recolored the cells for which a barcode was recovered in salmon (split A) or magenta (split B). Bar graphs demonstrating the raw cell distribution (left) and proportion (right) of barcoded cells across Seurat clusters in split A (salmon, top) or split B (magenta, bottom) as compared to GFP positive cells unable to be assigned a barcode (grey). (C) Heatmap showing normalized gene expression for the top 15 markers for Seurat clusters 1, 3, and 5 across all 17,599 cells. Mitochondrial markers are relatively overrepresented in cluster 5, whereas there are not good gene cluster markers for clusters 1 and 3.

Furthermore, the data in Figure 2 can be explained by the fact that cells are clonally contributing to specific cell types early in differentiation because of environmental bias and proliferative properties - certain cell types are derived by self-organising and local environmental self-reinforcing gene programs that could account for any clonal bias observed at the end of differentiation. I do not think there is sufficient evidence to suggest any clonal bias based on these barcoding studies.

We thank the reviewer for pointing out areas in which we should have been clearer about our claims. In the analyses represented by Figure 2 we determine that barcode clones—that is, cells descended from a common hiPS precursor—appear to be more clustered in expression state than what would have

been expected on random chance. However, our experiments cannot explain the mechanism underlying this property, which could be explained by environmental, proliferative, or intrinsic biases as well as influenced by self-organizing properties as the reviewer points out. We have now clarified these limitations more fully in the text.

From text (discussion):

Here we show that, during cardiac directed differentiation only a fraction of hiPS precursor cells contribute to the final differentiated population. It could be that all cells within particular barcode clones are all fated for extermination, or alternatively that cells are probabilistically selected out from within cells across all barcode clones equally. We cannot fully distinguish between these possibilities; however, our data suggest the latter scenario is more likely. **It is also possible that differences in proliferation, environmental cues, or self-organizing behavior underlie the biases in clonal behavior we observed.** We note that we recovered a smaller proportion of barcodes from differentiated cells in our single cell RNA-sequencing experiments than in our genomic DNA barcode sequencing and subsequently observed a smaller proportion of overlapping barcodes between parallel differentiations.

Figure 3-4 indicates that survival and starting gene expression state do not influence the contribution of iPSCs to differentiation potential. These experiments seem reasonably designed and interesting but the results are not surprising and do not significantly inform how we approach protocol development for differentiation.

We thank the reviewer for their positive comments about our experimental design. We agree that our results argue against specific intrinsic differences in differentiation potential in our experimental context. We do feel our study provides some value because in other instances, by contrast, there is significant intrinsic bias in differentiation (Weinreb et al. 2020). In that context, we feel that our study is a valuable contrasting contribution to the field.

For Figure 5 - the authors postulate that if cells displayed clonal division to give rise to restricted cell lineages then they would appear in clusters spatially in the well. I don't think this general concept is properly tested in this analysis - but instead it reflects how cells organise in a dish to differentiate - cell/cell interactions drive differentiation decisions. Evidence of clonal bias would only reflect the natural process of spatial bias in cell/cell environmental influences driving differentiation, not something intrinsic to the cells genetic predisposition. The authors say that "Less likely explanations for the observed marker patterns may be that similar cell types migrated towards each other to form these patches, or that unrelated, non-dividing cells in close proximity were all induced to become similar cell types". I disagree - I think there are natural cell micro-environmental factors that determine cell decisions and enable positional bias (clustering) in cell differentiation decisions. Therefore clonal bias is only a random consequence of what cues are received by a cell during differentiation and are completely independent of a cell autonomous differentiation bias. The data do not disprove this possibility and therefore the main conclusions of this data do not seem justified.

We agree with the reviewer completely on these points. Indeed, the primary conclusion of our manuscript is indeed that the microenvironmental factors are the dominant factors determining cell fate. We think that some of the confusion may come from terminology. Clonal bias, in our case, means that all cells of a clone tend towards the same fate. That may be due to microenvironmental factors or other factors noted above by the reviewer. Clonal bias does not, however, imply anything about biases due to long-lived variability in the intrinsic state of the cell before differentiation begins, which our "twin" experiments show does not exist. Thus, we concluded that it was likely either microenvironmental factors or short-lived intrinsic factors at the start of differentiation that determine cell fate. The spatial

analysis suggests that cells of a type tend to be near each other, suggesting that once a fate is locked in early in the differentiation process, that type remains over time. We have attempted to clarify these hypotheses in the main text as follows:

“...We confirmed this prediction from our RNA FISH results, where we observed that *TNNT2*-high, *ISL1*-high, and *EPCAM*-high cells all were found in patches on average of 15 or more cells, whereas *WT1*-high cells were found in patches of 6 cells, on average. **Therefore, these data raise the possibility that local early environmental cues may be important for the clustering of particular cell states we observed.** Less likely explanations for the observed marker patterns may be that similar cell types migrated towards each other to form these patches, or that unrelated, non-dividing cells in close proximity were all induced to become similar cell types, **potentially through self-organizing properties.** **Further work will be needed to conclusively eliminate such possibilities....”**

Reviewer #2: Here, Jiang et al., use elegant single cell lineage tracing methods to investigate cardiac directed differentiation of human induced pluripotent stem cells. Most directed differentiation methods lack efficiency and fidelity. However, the mechanisms underlying these current limitations have not been fully characterized. Thus, this study represents an important contribution to the stem cell field and an excellent demonstration of the value of genomic lineage tracing to a broader cell biology discipline. By associating differentiated cells with their early ancestors, the authors were able to demonstrate that differentiated cells that share an hiPS cell progenitor were more transcriptionally similar to each other than to other cells in the differentiated population. However, this transcriptional similarity was not observed when the same hiPS cells were differentiated in parallel. Together, with analysis of the spatial distribution of differentiating cells, these results suggest that cell fate decisions within this system are made shortly after cells are seeded. Overall, this is a sophisticated study that is performed with rigor. I have the following specific comments:

We thank the reviewer for their kind words about our study.

Major comments:

1. Overall, I think the analysis and statistical methods used are extremely elegant. I like the selection of the the Jensen-Shannon distance as a metric of similarity. I also think the experimental design - where two-thirds of the cells are differentiated in parallel, and one-third was DNA sequences determine the barcodes present in the initial population of hiPS cells to account for cell loss - is neat. However, one potentially weaker area of the analysis lies in whether it is sufficiently powered in some cases. For example, the authors note that 'cluster 3's behavior is likely artifactual and driven by its small size'. Can they provide some kind of formal test to support this? Furthermore, for some of the analyses, the selected clones consisted of only 5 cells in each split, which I think might be too few cells. Indeed, barcodes B1 and B2, displayed higher within-clone similarity across split differentiations, which might support the smaller clones generating a skewed outcome. We've run into similar issues with our own clonal analyses.

We thank the reviewer for their positive assessment of our analysis and experimental design. We also fully appreciate their point that there are some instances in which our ability to make inferences could be limited by our sample sizes. The reviewer has pointed out a few good examples, which we address here.

In the case of Cluster 3's behavior, we have clarified our assertion, which we agree was vague and not well supported. We have now done a far more comprehensive statistical analysis, finding that, statistically, one barcode was truly overrepresented in cluster 3 (barcode 35A), whereas the other barcodes were not (barcode 28A was overrepresented, but not significantly so). At the same time, we still hesitate to make statements about Cluster 3 because it has no clear markers and hence is harder to interpret biologically. We have now changed our wording to say:

“...As transcripts of the GFP transgene are captured by single cell RNA sequencing, we were able to recover and connect Rewind barcodes to individual cells through analysis of GFP transcript 10x sequencing reads (Supplementary Figures 2-3). We also profiled some of our GFP negative cell population and found the populations to be very similar to the GFP positive populations (Supplementary Figure 2), suggesting that the introduction of the barcode itself did not introduce major bias (there was some bias in clusters 1 and 3, but those clusters were difficult to interpret biologically due to a lack of good markers and hence were not subjected to further analysis (Supplementary Figure 3C). Through this process, we were able to both profile the expression state of individual differentiated cells and determine which differentiated cells were descended from the same barcoded hiPS cell progenitor. ...”

“...We found that the expression clusters with the strongest tendency to encompass all of a barcoded clone were clusters 4, 8, 3, and 12 (Supplementary Figure 4B). While, as aforementioned, cluster 3 is difficult to interpret biologically given its lack of clear expression markers (Supplementary Figure 3C), clusters 4, 8, and 12 represent putative cardiomyocytes, cardiac precursors, and epithelial cells, suggesting that these cell types may have stronger determination upon seeding for differentiation.”

These additional analyses are included in a modified Supplementary Figure 4 as follows:

Supplementary Figure 4

Supplementary figure 4. “Probability distribution” of barcoded clones across Seurat clusters. (A) Explanation of how we derived cluster probability distributions from raw cell cluster distribution for each barcoded clone, using barcode 31B as an example. (B) Heatmap demonstrating the normalized cell proportion across Seurat clusters for all barcodes in the analysis from Figure 2. Most clusters are associated with low normalized cell proportions (light purple) across barcodes, but clusters 4, 8, and 12 have more binary behavior (i.e. either high normalized cell proportion or very low normalized cell proportion). (C) Analysis determining whether barcode 28A and 35A cluster 3 overrepresentation as visualized in B is significant. Maintaining the organization provided by UMAP, we plotted all cells in the analysis (grey) and recolored cells corresponding to barcodes 28A (left) and 35A (right) in teal. Also for these barcodes, we plotted bar graphs for observed cluster probability distribution (teal) and the average random cluster probability distribution (grey). Finally, we plot histograms demonstrating the distribution of the number of cells that would land in cluster 3 by random chance as generated from 1000 random samplings of an matched number of cells in the analysis (i.e., sampling 25 cells for barcode 28A and 24 cells for barcode 35A, grey) as compared to the observed number of barcode 28A or 35A-labelled cells assigned to cluster 3 (teal vertical line). Cluster 3 behavior is statistically significant for some barcode clones (ex. barcode 35A), perhaps because few GFP positive cluster 3 clones are able to be assigned a barcode (see Supplementary Figure 3).

The reviewer has also made an excellent point about the number of cells used for our barcode analysis, and how the low number of cells we used may lead to some artifacts. To address this issue, we redid the analysis using thresholds of 7, 10, 15, and 20 cells, which yielded fewer barcodes in the analysis each time, but the qualitative trends remained the same; we included our findings in a modified Supplementary Figure 6 as follows:

Supplementary Figure 6

Supplementary figure 6. Jensen-Shannon distance captures known heritable predetermination of expression state. (A) Heatmap of pairwise Jensen-Shannon distances (smaller distance is darker in color) between split A and B cells associated with barcode clones with (from left to right) at least 7, 10, 15, or 20 cells per split. As demonstrated in Figure 4B, separated clones sharing a barcode (bolded outline along the diagonal) had Jensen-Shannon distances in the same range as clones labeled by distinct barcodes (off the diagonal). (B) Heatmap of pairwise Jensen-Shannon distances between cells associated with 2 barcodes in split A and the cells associated with the same 2 barcodes in split B from a dataset previously shown to have heritable predetermination of the final cell state (vemurafenib-treated melanoma cells). Separated clones sharing a barcode (bolded outline along the diagonal) had much smaller Jensen-Shannon distances than clones labeled by distinct barcodes (off the diagonal), demonstrating their similarity. This similarity is also visible in the comparison for each barcode of bar graphs for observed cluster probability distribution in split A (salmon) and the observed cluster probability distribution in split B (magenta). Cluster probability distributions are visually similar between separated clones and visually distinct across cells labeled by different barcodes.

The reviewer has also raised an excellent point that barcodes with the largest numbers of cells had different similarity than the smaller ones, which is a potential source of bias because the large number of smaller clones could be leading us to an erroneous conclusion. In the end, it is somewhat hard to say, because it also may be a legitimate biological difference between the larger vs. smaller clones, rather than just a sampling issue. We have now elaborated on these ambiguities in the text as follows:

“... For the 18 barcode clones consisting of at least 5 cells in each split, we calculated the pairwise Jensen-Shannon distances between all cluster probability distributions (Figure 4B). The Jensen-Shannon distance values between separated clones labeled by the same barcode (on the diagonal in Figure 4B) were in the same range as the distances between clones labeled by distinct barcodes (off

the diagonal in Figure 4B) for all but two barcodes. This was also borne out by visual inspection of the cluster probability distributions for barcoded clones, which generally appeared quite different between splits (Figure 4C). The overall trend towards dissimilar cluster probability distributions between clones that were differentiated separately suggests that final differentiated expression state is not broadly predetermined by intrinsic differences in hiPS cell precursors, at least at this time scale. The exceptions to this trend were barcodes B1 and B2, which displayed higher within-clone similarity across split differentiations; interestingly, these barcode clones contained many more cells than the other barcode clones in our analysis. Even if we were to take the B1 and B2 results without caveats, we could only interpret them to mean that a small subset of hiPS cells may be primed to take on certain ranges of final expression states. **We found a similar trend even when we analyzed clones with a minimum of 7, 10, 15 or 20 cells (Supplementary Figure 6A).** In contrast, when performing this analysis on barcoded vemurafenib-treated melanoma cells, a system with known heritable predetermination of the final cell state (Shaffer et al. 2020; Schuh et al. 2020; Emert et al. 2021), barcoded clones in one split were far more similar (i.e. had lower Jensen-Shannon distance values) to the related clones in the other split than to clones labeled by a distinct barcode (Supplementary Figure 6B). **An important consideration is that we analyzed many more clones with smaller numbers of cells; it is possible that the numerous small clones appear to have lower intrinsic potential due to low sampling and are masking a more obvious intrinsic potential in large clones. It may also be that the differences in intrinsic potential may reflect biological differences between clones; further experiments with more sampling would be required to distinguish these possibilities...."**

2. It would be helpful to include additional metrics, such as the percentage of cells captured by flow cytometry. Additionally, the authors claim that the GFP negative cell population was transcriptionally indistinguishable - it would be helpful to include these results.

We thank the reviewer for this suggestion and have included these results in a new Supplementary Figure 2 (see below). These results provide justification for the claim that the GFP negative cell population is transcriptionally very similar to the GFP positive one. We agree that the inclusion of these key data help strengthen our conclusions.

Supplementary Figure 2

Supplementary figure 2. GFP positive and negative sorted cells are largely transcriptionally indistinguishable. (A) We sorted cells from both splits A and B by GFP expression using FACS. Dots represent individual cells. Plots were constructed using the final 50,000 sorted cells from each sample. (B) Maintaining the organization provided by UMAP, we plotted all sequenced GFP negative sorted cells in grey and all sequenced GFP positive sorted cells in green. Bar graphs demonstrating the distribution of GFP positive and negative cells across Seurat clusters, colored the same way. Note that we only sequenced a portion of the GFP negative population, taken from the sorted split B GFP negative cells.

3. Overall, there are several parallels between this study and reports in the cell reprogramming arena. For example, work from Rudy Jaenisch's lab rules out the existence of 'elite' cell populations that are primed to reprogram and instead suggests that cells transition through 'privileged' states in which they are more likely to reprogram (Hanna et al., Nature, 2009). Similarly, in our own lineage tracing of reprogramming, we showed that sisters within the same biological replicate shared the same reprogramming outcomes, established early in the fate conversion process. However, sisters split across parallel replicates did not share reprogramming outcomes (Bidy et al., Nature, 2009). However, the authors' experimental design in this current study to incorporate DNA sequencing to account for absent sisters is more elegant. Together, I think these studies point to the importance of studying early events in differentiation/reprogramming and the power of lineage tracing tools to enable these investigations.

The reviewer has pointed out several critical references and ideas that provide important context for our work. We have now elaborated on these ideas in the discussion.

“...differentiated cells may have been more affected than experiments where we had more initial barcodes (i.e. higher MOI).

Our results have several parallels with other reprogramming and differentiation systems that have similarly found that “elite” primed cells can be short lived or functionally absent within certain progenitor populations. Hanna et al. showed that reprogramming happens in transient populations (Hanna et al. 2009), and Bidy et al. use lineage tracing to show that reprogramming outcome was

different even between sibling cells (Bidy et al. 2018). It may be that some of these cell states have a very short persistence time, or that microenvironmental factors are dominant.

We focused on cell type determination during cardiac directed differentiation of hiPS cells because the protocols are well-characterized...”

Minor comments:

1. Although the barcode library is presumably highly complex, it would be helpful for the authors to include details of this in the methods and the expected homoplasmy given the sizes of the populations labeled.

We thank the reviewer for this suggestion and have elaborated on this in the Methods section, with inclusion of a relevant diagram in Supplementary Figure 3A (see above). We first estimated our library sizes in Emert et al. 2021 (Supplementary Figure 2 of that paper), calculating pairwise Hamming and Levenshtein sequence distances for 20 million randomly sampled pairs of barcodes extracted from separate transduction, concluding that our library was “sufficiently diverse to label 100,000s of cells with over 99% receiving unique barcodes” (Emert et al. 2021). Having remade new libraries using the same method, we also sequenced the outcome of multiple independent transductions of the same library pool, looking for overlap. Using mark-recapture analysis, we estimate our library size to be >45 million unique barcodes, which we think is sufficient to avoid many duplicates (Goyal et al. 2021).

“...We pooled the plasmids from the 9 separate cultures in equal amounts by weight before packaging into lentivirus. We estimated our library complexity as described elsewhere (Goyal et al. 2021). Briefly, we sequenced three independent transductions in WM989 A6-G3 melanoma cells and took note of the total and pairwise overlapping extracted barcodes. Using the mark-recapture analysis formula we estimate our barcode diversity from these three transductions to be between 48.9 and 63.3 million barcodes....”

2. With respect to the visualization of clones, light pink on grey is difficult to distinguish. Perhaps a different symbol could be used, maybe an x?

We thank the reviewer for this suggestion and after trying different colors and symbols have amended the light pink clones to be a hopefully easier to discern salmon color instead.

Signed, Samantha Morris, Washington University.

Second round of review

Reviewer 1

The authors have carefully revised the manuscript to provide important clarifications to the interpretation of results. Overall I am pleased with these changes which have addressed my core concerns about the study design and analysis.

Reviewer 2

The authors have addressed all my previous concerns.